# Two Heads are Better than One: Towards Better Adversarial Robustness by Combining Transduction and Rejection

## Abstract

Both transduction and rejection have emerged as important techniques for defending against adversarial perturbations. A recent work by Tramer (2022) showed that, in the rejection-only case (no transduction), a strong rejection-solution can be turned into a strong (but computationally inefficient) non-rejection solution. This detector-to-classifier reduction has been mostly applied to give evidence that certain claims of strong selective-model solutions are susceptible, leaving the benefits of rejection unclear. On the other hand, a recent work by Goldwasser et al. (2020) showed that *rejection combined with transduction* can give *provable* guarantees (for certain problems) that cannot be achieved otherwise. Nevertheless, under recent strong adversarial attacks (GMSA (Chen et al., 2022), which has been shown to be much more effective than AutoAttack against transduction), Goldwasser et al.'s work was shown to have low performance in a practical deep-learning setting. In this paper, we take a step towards realizing the promise of transduction+rejection in more realistic scenarios. Theoretically, we show that a novel application of Tramèr's classifier-to-detector technique in the transductive setting can give significantly improved sample-complexity for robust generalization. While our theoretical construction is computationally inefficient, it guides us to identify an efficient transductive algorithm to learn a selective model. Extensive experiments using state of the art attacks (AutoAttack, GMSA) show that our solutions provide significantly better robust accuracy.

## 1 Introduction

A recent line of research (Goldwasser et al., 2020; Montasser et al., 2021; Goodfellow, 2019; Wang et al., 2021; Wu et al., 2020) has investigated augmenting models with *transduction* or *rejection* to defend against adversarial perturbations. However, the results of leveraging these new options have been mixed. For example, a recent work by Tramer (2022) gives an equivalence between classification-only and classification-with-rejection; the major application of the author's results has been to provide bounds on the performance of defenses with rejection, which can be used to show that the robustness of defenses with rejection may be lower than the authors originally claimed, casting doubt on the benefits of rejection.

On the other hand, some recent work in theory has demonstrated that *transduction*, that is leveraging the unlabeled test-time input for learning the model, may have significant impact on adversarial robustness. Specifically, Montasser et al. (2021) studied the setting of transduction (without rejection), and show that robust learning with transduction allows for significant improvemnents in sample complexity, reducing dependency on VC dimension from exponential to linear; however, this comes at the cost of significantly greater assumptions on the data ($\text{OPT}_{\mathcal{U}^2}$ for the realizable case rather than the $\text{OPT}_{\mathcal{U}}$ of the inductive setting [1]). Goldwasser et al. (2020) studied transduction and rejection, and show even more surprising results, not achievable with transduction or rejection alone. However, one prominent limitation of these works seems to be that none has yet resulted in practical robust learning mechanisms in the deep learning setting typically considered.

---

[1]The optimal robust risk is $\text{OPT}_{\mathcal{U}} = \inf_{h \in \mathcal{H}} \Pr_{(x,y) \sim \mathcal{D}} [\exists z \in \mathcal{U}(x) : h(z) \neq y]$.

| | Realizable | | | Agnostic Generalization Bound |
| --- | --- | --- | --- | --- |
| | Soundness Condition | Completeness Condition | Generalization Bound | |
| Induction (Montasser et al., 2019) | $\mathrm{OPT}_{\mathcal{U}} = 0$ | $\mathrm{OPT}_{\mathcal{U}} = 0$ | $O\left(\frac{2^{\mathrm{VC}(\mathcal{H})}\log(n)+\log(1/\delta)}{n}\right)$ | $\mathrm{OPT}_{\mathcal{U}} + O\left(\sqrt{\frac{2^{\mathrm{VC}(\mathcal{H})}+\log(1/\delta)}{n}}\right)$ |
| Transduction (Montasser et al., 2021) | $\mathrm{OPT}_{\mathcal{U}^2} = 0$ | $\mathrm{OPT}_{\mathcal{U}^2} = 0$ | $O\left(\frac{\mathrm{VC}(\mathcal{H})\log(n)+\log(1/\delta)}{n}\right)$ | $2\mathrm{OPT}_{\mathcal{U}^2} + O\left(\sqrt{\frac{\mathrm{VC}(\mathcal{H})+\log(1/\delta)}{n}}\right)$ |
| Rejection (Theorem A.2, A.6) | $\mathrm{OPT}^{\mathrm{rej}}_{\mathcal{U}} = 0$ | $\mathrm{OPT}^{\mathrm{rej}}_{\mathcal{U}} = 0$ | $O\left(\frac{2^{\mathrm{VC}(T(\mathcal{H}))}\log(n)+\log(1/\delta)}{n}\right)$ | $\mathrm{OPT}^{\mathrm{rej}}_{\mathcal{U}} + O\left(\sqrt{\frac{2^{\mathrm{VC}(T(\mathcal{H}))}+\log(1/\delta)}{n}}\right)$ |
| Transduction+Rejection (Goldwasser et al., 2020) | $\mathrm{OPT}_{\mathcal{U}} = 0$ | $\mathrm{OPT}_{\mathcal{U}} = 0$ | $O\left(\sqrt{\frac{\mathrm{VC}(\mathcal{H})\log(n)}{n}} + \frac{\log(1/\delta)}{n}\right)$ | $2\,\mathrm{OPT}_{\mathcal{U}} + 2\sqrt{2\,\mathrm{OPT}_{\mathrm{I}}} + O\left(\sqrt{\frac{\mathrm{VC}(\mathcal{H})\log n+\log(1/\delta)}{n}}\right)$ |
| Transduction+Rejection (Theorem 4.1, A.12) | $\mathrm{OPT}_{\mathcal{U}^{\ell 2/3}} = 0$ | $\mathrm{OPT}_{\mathcal{U}^2} = 0$ | $O\left(\frac{\mathrm{VC}(\mathcal{H})\log(n)+\log(1/\delta)}{n}\right)$ | $2\mathrm{OPT}_{\mathcal{U}^{\ell 2/3}} + O\left(\sqrt{\frac{\mathrm{VC}(\mathcal{H})+\log(1/\delta)}{n}}\right)$ |

Table 1: **Summary of generalization bounds for the four settings**. Compared to transduction alone and Goldwasser et al. (2020), our defense weakens the necessary conditions in the realizable case and improves the asymptotic error in the agnostic case. Compared to induction and rejection alone, sample complexity has a linear rather than exponential dependence on the VC dimension. Compared to Goldwasser et al. (2020), the dependence on the error bound $\epsilon$ improves from inverse quadratic to inverse linear in the realizable case.

Specifically, compared to Goldwasser et al. (2020), which considered arbitrary perturbations, we focus on the classic and practical scenario of bounded perturbations for deep learning. Somewhat surprisingly, we show that a novel application of Tramèr's classifier-to-detector technique in the transductive setting can give significantly improved sample-complexity for robust generalization, noting that bounded perturbations are critical for the construction to work. To obtain these improvements, we do not require stronger assumptions on the data, as with Montasser et al. (2021); in the realizable case, we only need to assume $\mathrm{OPT}_{\mathcal{U}^{\ell 2/3}} = 0$, which is even better than the $\mathrm{OPT}_{\mathcal{U}} = 0$ assumption in the inductive case. Table 1 gives more details; the notation is described in Section 3. Our results give a first constructive application of Tramèr's classifier-to-detector reduction which leads to *improved* sample complexity.

While our theoretical construction is computationally inefficient due to the use of Tramèr's reduction, it guides us to identify a practical transductive algorithm for learning a robust selective model. In addition, we present an objective for general adaptive attacks targeting selective classifiers based on our algorithm. Our transductive defense algorithm gives strong empirical performance on image classification tasks, both against our adaptive attack and against existing state-of-the-art attacks such as AutoAttack and standard GMSA. We obtain 81.6% and 57.9% transductive robust accuracy with rejection on CIFAR-10 and CIFAR-100, respectively, a significant improvement on the current state-of-the-art result of 71.1% and 42.7% (Peng et al., 2023; Wang et al., 2023; Croce et al., 2020) for robust accuracy up to the perturbation considered ($l_\infty$ with budget $\epsilon = 8/255$).

The rest of the paper is organized as follows. Section 2 reviews main related work, and Section 3 presents some necessary background. We develop our theory results in Section 4. Guided by our theory, Section 5 develops a practical robust learning algorithm, leveraging both transduction and rejection. We provide systematic experiments in Section 6, and conclude in Section 7.

## 2 RELATED WORK

In recent years, there have been extensive studies on adversarial robustness in the traditional inductive learning setting, where the model is fixed during the evaluation phase (Carlini & Wagner, 2017; Goodfellow et al., 2015; Moosavi-Dezfooli et al., 2016). Most popular and effective methods are adversarial training, such as PGD (Madry et al., 2018), TRADES (Zhang et al., 2019). These methods are effective against adversaries on small dataset like MNIST, but still ineffective on complex dataset like CIFAR-10 or ImageNet (Croce et al., 2020). Defenses beyond adversarial training have been proposed but most are broken by strong adaptive attacks (Croce & Hein, 2020; Tramer et al., 2020).

To break this robust bottleneck, recent work has proposed alternative settings with relaxed yet realistic assumptions, particularly by allowing rejection and transduction. In robust learning with rejection (a.k.a., abstain), we allow rejection of adversarial examples instead of correctly classifying all of them (Tramer, 2022). Variants of adversarial training with rejection option have been considered (Laidlaw & Feizi, 2019; Pang et al., 2022; Chen et al., 2021; Kato et al., 2020; Sotgiu et al., 2020; He et al., 2022), including generalizations to unseen attacks (Stutz et al., 2020) and to certified robustness (Sheikholeslami et al., 2020; Baharlouei et al., 2022; Sheikholeslami et al., 2022).

(Tramer, 2022) proves an equivalence between robust learning with rejection and standard robust learning in the inductive setting and shows that the evaluation of past defenses with rejection was unreliable.

The other approach is to define an alternative notion of adversarial robustness via transductive learning, i.e. "dynamically" ensuring robustness on the particular given test samples rather than on the whole distribution. Similar settings have been studied but under the view of "test-time defense" or "dynamic defense" (Goodfellow, 2019; Wang et al., 2021; Wu et al., 2020). (Goldwasser et al., 2020) is the first paper to formalize transductive learning for robust learning, and the first to consider transduction+rejection. It considers general adversaries on test data and presents novel theoretical guarantees. (Chen et al., 2022) formally defines the notion of transductive robustness as a maximin problem and presents a principled adaptive attack, GMSA. (Montasser et al., 2021) discusses robust transductive learning against bounded perturbation from a learning theory perspective and obtains corresponding sample complexity.

## 3 Preliminaries

| | Robust Error | Robust Error (with Rejection) |
|---|---|---|
| Inductive | $\mathrm{err}_{\mathcal{U}}(h; x, y) := \sup_{z \in \mathcal{U}(x)} \mathbb{1}\{h(z) \neq y\}$ | $\mathrm{err}_{\mathcal{U}}^{\mathrm{rej}}(h; x, y) := \sup_{z \in \mathcal{U}(x)} \mathbb{1}\{h(z) \notin \{y, \bot\} \vee h(x) \neq y\}$ |
| Transductive | $\mathrm{err}_{\mathcal{U}}(h; \boldsymbol{x}, \boldsymbol{y}, \tilde{\boldsymbol{z}}, \tilde{\boldsymbol{y}}) := \frac{1}{m} \sum_{i=1}^{m} \mathbb{1}\{h(\tilde{z}_i) \neq \tilde{y}_i\}$ | $\mathrm{err}_{\mathcal{U}}^{\mathrm{rej}}(h; \boldsymbol{x}, \boldsymbol{y}, \tilde{\boldsymbol{x}}, \tilde{\boldsymbol{z}}, \tilde{\boldsymbol{y}}) := \frac{1}{m} \sum_{i=1}^{m} \mathbb{1} \left\{ \begin{array}{c} (h(\tilde{z}_i) \notin \{\tilde{y}_i\} \wedge \tilde{z}_i = \tilde{x}_i) \\ \vee (h(\tilde{z}_i) \notin \{\tilde{y}_i, \bot\} \wedge \tilde{z}_i \neq \tilde{x}_i) \end{array} \right\}$ |

Table 2: Summary of the robust error in all settings. Note that transductive error of the learner $\mathbb{A}$ is the corresponding notion of error where $h = \mathbb{A}(\boldsymbol{x}, \boldsymbol{y}, \tilde{\boldsymbol{z}})$.

Let $\mathcal{X}$ denote the input space, $\mathcal{Y}$ the label space, $\mathcal{D}$ the clean data distribution over $\mathcal{X} \times \mathcal{Y}$. We will assume binary classification for our theoretical analysis: $\mathcal{Y} = \{\pm 1\}$. Let $\mathcal{U}(x)$ denote the set of possible perturbations of an input $x$, e.g., for $\ell_p$ norm perturbation of budget $\epsilon$, $\mathcal{U}$ is the $\ell_p$ ball of radius $\epsilon$: $\mathcal{U}(x) = \{z : \|z - x\|_p \leq \epsilon\}$. We assume $\mathcal{U}$ satisfies $\forall x \in \mathcal{X}, x \in \mathcal{U}(x)$; essentially all interesting perturbations satisfy this. Let $\mathcal{U}^2(x) := \{z : \exists t \in \mathcal{U}(x), \text{such that } z \in \mathcal{U}(t)\}$, and $\mathcal{U}^{-1}(x) := \{z : x \in \mathcal{U}(z)\}$. If a perturbation set $\Lambda$ satisfies $\Lambda^2 = \mathcal{U}$, then we say $\Lambda = \mathcal{U}^{1/2}$; $\mathcal{U}^{-1/2} = (\mathcal{U}^{-1})^{1/2}$. When $\mathcal{U}$ is the $\ell_p$ ball of radius $\epsilon$, $\mathcal{U}^2$ is that of radius $2\epsilon$, $\mathcal{U}^{-1} = \mathcal{U}$, and $\mathcal{U}^{1/2}$ is that of radius $\epsilon/2$; we define $\mathcal{U}^3$, $\mathcal{U}^{1/3}$, and $\mathcal{U}^{-1/3}$ similarly.

All learners are provided with $n$ i.i.d. training samples [2] $(\boldsymbol{x}, \boldsymbol{y}) = (x_i, y_i)_{i=1}^n \sim \mathcal{D}^n$. There are $m$ i.i.d. test samples $(\tilde{\boldsymbol{x}}, \tilde{\boldsymbol{y}}) \sim \mathcal{D}^m$, and the adversary can perturb $\tilde{\boldsymbol{x}}$ to $\tilde{\boldsymbol{z}} \in \mathcal{U}(\tilde{\boldsymbol{x}})$. We describe the main settings below; the corresponding notions of error are in Table 2. For each setting, we define risk as the expected worst-case error up to the perturbation $\mathcal{U}$, and empirical risk similarly.

**Induction.** In the traditional robust classification setting (e.g., Madry et al. (2018)); also called the inductive setting or simply induction), the learning algorithm (the defender) is given training set $(\boldsymbol{x}, \boldsymbol{y})$, learns a classifier $h : \mathcal{X} \mapsto \mathcal{Y}$ from some hypothesis class $\mathcal{H}$.

**Rejection.** In the setting of robust classification with rejection, the classifier has the extra power of abstaining (i.e., outputting a rejection option denoted by $\bot$), and furthermore, rejecting a perturbed input does not incur an error. The learning algorithm is given training set $(\boldsymbol{x}, \boldsymbol{y})$ and learns a *selective classifier* $h : \mathcal{X} \mapsto \mathcal{Y} \cup \{\bot\}$ from some hypothesis class $\mathcal{H}$. An error occurs only when $h$ rejects a clean input, or accepts and misclassifies. We define additionally $\mathrm{OPT}_{\mathcal{U}}^{\mathrm{rej}} := \inf_{h \in \mathcal{H}} \mathrm{R}_{\mathcal{U}}^{\mathrm{rej}}(h; \mathcal{D})$.

**Transduction.** In the setting of robust classification with transduction (e.g., Montasser et al. (2021)), the learning algorithm (the transductive learner) has access to the unlabeled test input data; the goal is to predict labels only for these given test inputs (a transductive learner need not generalize). The learner $\mathbb{A}$ is given the training data $(\boldsymbol{x}, \boldsymbol{y})$ and the (potentially perturbed) test inputs $\tilde{\boldsymbol{z}}$, and outputs $m$ labels $h(\tilde{\boldsymbol{z}}) = (h(\tilde{z}_i))_{i=1}^m$ as predictions for $\tilde{\boldsymbol{z}}$. That is, the learner is a mapping $\mathbb{A} : (\mathcal{X} \times \mathcal{Y})^n \times \mathcal{X}^m \mapsto \mathcal{Y}^m$. A special case is when $\mathbb{A}$ learns a classifier $h$ and use it to label $\tilde{\boldsymbol{z}}$; the labels are also denoted as $h(\tilde{\boldsymbol{z}})$.

---

[2] Here $\boldsymbol{x} = (x_i)_{i=1}^n$ and similarly with $\boldsymbol{y}, \tilde{\boldsymbol{x}}, \tilde{\boldsymbol{y}}$, etc. We will also overload the notation $\mathcal{U}$, e.g., $\mathcal{U}(\boldsymbol{x}) := \{\boldsymbol{u} \in \mathcal{X}^n : u_i \in \mathcal{U}(x_i)\}$.

**Our setting: Transduction+Rejection.** A transductive learner for selective classifiers $\mathbb{A}$ is given $(x, y, \tilde{z})$, and outputs rejection or a label for each input in $\tilde{z}$. That is, the learner is a mapping $\mathbb{A} : (\mathcal{X} \times \mathcal{Y})^n \times \mathcal{X}^m \mapsto (\mathcal{Y} \cup \{\perp\})^m$. An error occurs when it rejects a clean test input or accepts and misclassifies.

## 4 THEORETICAL ANALYSIS

In this section, we present the theorem statements and proof sketches for the realizable case in the setting with transduction and rejection. The proof details and results for the agnostic case and the setting with rejection alone are in Appendix A.

For comparison with existing results in the inductive-only and transduction-only settings (Montasser et al., 2019; 2021), we follow their setup: assume there exists a classifier (without rejection) with 0 robust error from a hypothesis class $\mathcal{H}$ of VC-dimension $\mathrm{VC}(\mathcal{H})$, and the learner constructs a selective classifier for labeling the test inputs (or constructs a set of selective classifiers and uses any of them for labeling). The goal is to design a learner with a small robust error.

**Theorem 4.1.** *For any $n \in \mathbb{N}$, $\delta > 0$, hypothesis class $\mathcal{H}$ of classifiers without rejection, perturbation set $\mathcal{U}$ such that $\mathcal{U} = \mathcal{U}^{-1}$ and $\mathcal{U}^{1/3}$ exists, and distribution $\mathcal{D}$ over $\mathcal{X} \times \mathcal{Y}$ satisfying $\mathrm{OPT}_{\mathcal{U}^{2/3}} = 0$, there exists a transductive learner $\mathbb{A}$ that constructs a set of selective classifiers $\Delta$ s.t. the following is true: with probability $\geq 1 - \delta$ over $(x, y) \sim \mathcal{D}^n$, $(\tilde{x}, \tilde{y}) \sim \mathcal{D}^n$, we have that for any $\tilde{z} \in \mathcal{U}(\tilde{x})$, if $\Delta \neq \emptyset$, then for any $h \in \Delta$,* [3]

$$\mathrm{err}_{\mathcal{U}}^{\mathrm{rej}}(h; x, y, \tilde{x}, \tilde{z}, \tilde{y}) \leq \frac{\mathrm{VC}(\mathcal{H}) \log(2n) + \log(1/\delta)}{n}.$$

For $\mathcal{U}$ satisfying our conditions (including $l_p$ balls), we obtain a stronger guarantee than those using only transduction or only rejection. First, compared to the guarantee for transduction without rejection (Montasser et al., 2021) (see Table 1), our result requires weaker assumptions on the data: we need $\mathrm{OPT}_{\mathcal{U}^{2/3}} = 0$ rather than $\mathrm{OPT}_{\mathcal{U}^2} = 0$. For example, consider the $\ell_p$ norm perturbation: $\mathcal{U}(x) = \{z : \|z - x\|_p \leq \epsilon\}$. Transduction alone requires that there exists a classifier with 0 robust error up to the perturbation $\mathcal{U}^2$, i.e. up to an $\ell_p$ norm perturbation of adversarial budget $2\epsilon$. In contrast, our result shows that using both transduction and rejection only requires there exists a classifier with 0 robust error up to perturbation $\mathcal{U}^{2/3}$, corresponding to adversarial budget of $2\epsilon/3$. Equivalently, for a data distribution with a margin $2\epsilon$, transduction without rejection can only handle adversarial perturbations with budget $\epsilon$, while combining transduction and rejection can handle adversarial perturbations with budget $3\epsilon$, tolerating three times the adversarial magnitude. Second, compared to rejection only (see Table 1), this bound has a linear sample complexity rather than exponential. Therefore, combining transduction and rejection has the benefits of both techniques.

Furthermore, note that the result bounds the rate of incorrect rejections as well, i.e. the rate of rejections on clean data, with the same bound as a direct consequence of the definition of robust error under transduction and rejection. However, the result, while potentially very strong, comes with the caveat that the defense is not guaranteed to find a nonempty $\Delta$ (i.e., the defense is sound but may not be complete) under conditions weaker than $\mathrm{OPT}_{\mathcal{U}^2} = 0$; by Lemma A.14 in Appendix A.3, $\Delta$ is guaranteed to be nonempty, and hence we have completeness, under the same conditions as transduction alone. Hence, the result is strictly stronger than the result for transduction alone (Montasser et al., 2021).

Consider an adversarial budget $\epsilon$, and suppose $\tilde{z}$ is the given potentially perturbed test input and $\tilde{x}$ is the corresponding clean test input. To obtain the guarantee, we need to find a model which is $\epsilon/3$-robust at $q = \tilde{x} + (\tilde{z} - \tilde{x})/3$. Such a model always exists when $\mathrm{OPT}_{\mathcal{U}^{2/3}} = 0$. However, given only $\tilde{z}$ without knowing $q$ or $\tilde{x}$, our algorithm finds a model $\epsilon/3$-robust at every perturbation within $2\epsilon/3$ of $\tilde{z}$ and thus $\Delta$ may be empty.

While weaker conditions don't guarantee that we find a model satisfying the conditions, the result still provides intuition for the success of our derived empirical defense. For typical data distributions and hypothesis classes, it might be expected that, if we fail to find a $\epsilon$-robust hypothesis at the

---

[3] Note that $\Delta$ is a function of $x$, $y$, and $\tilde{z}$, so this is more precisely a bound of $\sup_{\tilde{z} \in \mathcal{U}(\tilde{x}), h \in \mathbb{A}(x, y, \tilde{z})} \mathrm{err}_{\mathcal{U}}^{\mathrm{rej}}(h; x, y, \tilde{x}, \tilde{z}, \tilde{y})$.

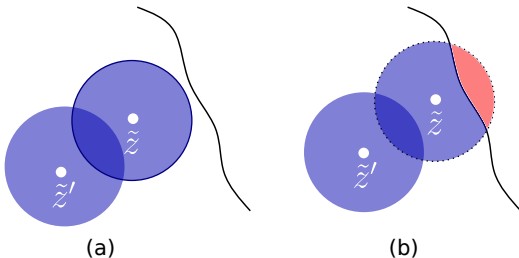

Figure 1: (a) $h$ is $\epsilon/3$-robust at $\tilde{z}$; $\hat{h}$ correctly classifies $\tilde{z}$. (b) $h$ is not $\epsilon/3$-robust at $\tilde{z}$; $\hat{h}$ rejects $\tilde{z}$.

fully-perturbed data, we will nevertheless be more likely to find a model which is robust nearer the clean data distribution (i.e. where the condition is required by the theory) rather than further away. Determining conditions for this is an interesting direction for future research. Note that such conditions do exist: in Appendix A.3 we present a distribution $\mathcal{D}$, hypothesis class $\mathcal{H}$, and perturbation $\mathcal{U}$ for which $\Delta$ is guaranteed to be nonempty and the error bound above applies, but where transduction has a minimum asymptotic error of $1/2$.

**Proof Sketch.** For intuition, think of $\mathcal{U}$ as the $\ell_p$ norm perturbation with adversarial budget $\epsilon$. We omit technical details; see Appendix A.3 for the complete proof. Consider some clean training set $\boldsymbol{x}, \boldsymbol{y}$, clean test set $\tilde{\boldsymbol{x}}, \tilde{\boldsymbol{y}}$, with perturbed test data $\tilde{\boldsymbol{z}}$ with $\tilde{z}_i$ within $\epsilon$ of $\tilde{x}_i$. Let $\tilde{\boldsymbol{z}}' = \tilde{\boldsymbol{x}} + (\tilde{\boldsymbol{z}} - \tilde{\boldsymbol{x}})/3$ be the intermediate perturbation a third of the way between $\tilde{\boldsymbol{x}}$ and $\tilde{\boldsymbol{z}}$.

First, following Montasser et al. (2021), define the set of robust hypotheses $\Delta_{\mathcal{H}}^{\mathcal{U}^{1/3}}(\boldsymbol{x}, \boldsymbol{y}, \tilde{\boldsymbol{z}}')$ as $\Delta_{\mathcal{H}}^{\mathcal{U}^{1/3}}(\boldsymbol{x}, \boldsymbol{y}, \tilde{\boldsymbol{z}}') = \{R_{\mathcal{U}^{1/3}}(h; \boldsymbol{x}, \boldsymbol{y}) = 0 \wedge R_{\mathcal{U}^{1/3}}(h; \tilde{\boldsymbol{z}}') = 0\}$. That is, we find those classifiers that satisfy: (1) they are $\epsilon/3$-robustly correct (i.e., correct and robust to perturbations of budget $\epsilon/3$) on the training data $(\boldsymbol{x}, \boldsymbol{y})$; (2) they have $\epsilon/3$ margin on the intermediate perturbations $\tilde{\boldsymbol{z}}'$ (i.e., have the same prediction for all perturbations of budget $\epsilon/3$). This then guarantees, as shown in Montasser et al. (2021), that with high probability, for any $h \in \Delta_{\mathcal{H}}^{\mathcal{U}^{1/3}}(\boldsymbol{x}, \boldsymbol{y}, \tilde{\boldsymbol{z}}')$ the robust error facing perturbation of budget $\epsilon/3$ is bounded by $\frac{\text{VC}(\mathcal{H})\log(2n)+\log(1/\delta)}{n}$ if $\text{OPT}_{\mathcal{U}^{2/3}} = 0$.

Next, following Tramer (2022), define a transformation $F_{\mathcal{U}^{1/3}}$ that maps a classifier without rejection, $h$, to the selective classifier $\hat{h} = F_{\mathcal{U}^{1/3}}(h)$: $\hat{h}(x) = \begin{cases} h(x) & \text{if } \forall x' \in \mathcal{U}^{-1/3}(x), h(x') = h(x) \\ \perp & \text{otherwise} \end{cases}$. That is, $\hat{h}$ rejects $x$ if it is within $\epsilon/3$ from $h$'s decision boundary, otherwise accepts and predicts $h(x)$.

Now, consider a clean test sample $(\tilde{x}, \tilde{y})$ and $\tilde{x}$'s adversarial perturbation $\tilde{z}$. Define an intermediate perturbation $\tilde{z}' = \tilde{x} + (\tilde{z} - \tilde{x})/3$. We will show that if $h$ is correct at $\tilde{z}'$, then $\hat{h}$ makes no error at $\tilde{z}$.

If $\tilde{z} = \tilde{x}$, then $\tilde{z}' = \tilde{x} = \tilde{z}$. Since $h$ is $\epsilon/3$-robust at $\tilde{z}'$, $h(\tilde{z}) = h(\tilde{z}') = \tilde{y}$ and so $\hat{h}(\tilde{z}) = \tilde{y}$ which is correct. Otherwise, we need to consider two cases: **(a)** $h$ is $\epsilon/3$-robust at $\tilde{z}$; **(b)** $h$ is not. See visualization in Figure 1. In both cases, the $\epsilon/3$-balls about $\tilde{z}$ and $\tilde{z}'$ intersect. Let $\tilde{z}''$ be some point in the intersection. Since $h$ is $\epsilon/3$-robust at $\tilde{z}'$, $h(\tilde{z}'') = h(\tilde{z}') = \tilde{y}$. Now, in case (a) where $h$ is $\epsilon/3$-robust at $\tilde{z}$, $h(\tilde{z}) = h(\tilde{z}'') = \tilde{y}$, which is correct. In case (b) where $h$ is not $\epsilon/3$-robust at $\tilde{z}$, $\hat{h}$ rejects $\tilde{z}$ and makes no error.

Hence if $h$ is correct at $\tilde{z}'$, then $\hat{h}$ makes no error at $\tilde{z}$. So the error bound for $h$ implies the desired error bound for any $\hat{h}$ in the set $\Delta' = \left\{\hat{h} = F_{\mathcal{U}^{1/3}}(h) : h \in \Delta_{\mathcal{H}}^{\mathcal{U}^{1/3}}(\boldsymbol{x}, \boldsymbol{y}, \tilde{\boldsymbol{z}}')\right\}$.

As we have access only to the adversarial test data $\tilde{\boldsymbol{z}}$, to ensure $\epsilon/3$-robustness at the unknown $\tilde{\boldsymbol{z}}'$, we need to ensure $\epsilon$-robustness at $\tilde{\boldsymbol{z}}$. Let $\Delta'' := \cup\left\{\hat{h} = F_{\mathcal{U}^{1/3}}(h) : h \in \bigcap_{\tilde{\boldsymbol{z}}' \in \mathcal{U}^{-2/3}(\tilde{\boldsymbol{z}})} \Delta_{\mathcal{H}}^{\mathcal{U}^{1/3}}(\boldsymbol{x}, \boldsymbol{y}, \tilde{\boldsymbol{z}}')\right\}$ and let $\hat{\Delta} = \bigcup_{\tilde{\boldsymbol{z}}' \in \mathcal{U}^{-2/3}(\tilde{\boldsymbol{z}})} \Delta_{\mathcal{H}}^{\mathcal{U}^{1/3}}(\boldsymbol{x}, \boldsymbol{y}, \tilde{\boldsymbol{z}}')$. By the above, as $\Delta'' \subseteq \Delta'$, any $\hat{h}$ in $\Delta''$ achieves the desired bound. If $|\hat{\Delta}| = 1$, then $|\Delta'| = 1$ and as $\Delta' \subseteq \hat{\Delta}$, $\hat{\Delta} = \Delta'$ and so any $\hat{h}$ in $\Delta'' \cup \hat{\Delta}$ likewise achieves the bound.

Hence, if we let $\Delta = \begin{cases} \Delta'' \cup \hat{\Delta} & |\hat{\Delta}| = 1, \\ \Delta'' & \text{otherwise} \end{cases}$, we obtain the theorem statement.

**Remark:** More direct approaches may seem possible, but have surprising pitfalls. At first glance, this approach may seem less natural than simply applying the analysis of Montasser et al. (2021) to a potential $\tilde{z}' \in \mathcal{U}^{1/2}(\tilde{x})$ with the condition of OPT$_{\mathcal{U}}$, obtaining a $\mathcal{U}^{1/2}$-robust classifier $h'$, and deriving an $\epsilon$-robust selective classifier by the transformation F$_{\mathcal{U}^{1/2}}$. While this seems possible at first, as Tramer (2022) shows that applying this transformation results in doubled robustness, this isn't possible in this situation, as $h'$ is only guaranteed to be $\mathcal{U}^{1/2}$-robust at $\tilde{z}'$, not at *every* $\epsilon/2$ perturbation of $\tilde{x}$ as needed by the analysis. Similarly, it might seem possible to obtain an $\epsilon/2$-robust classifier at $\tilde{z}$ using Montasser et al. (2021), and derive the desired $\epsilon$-robust classifier from F$_{\mathcal{U}^{1/2}}$; this, however, requires the condition OPT$_{\mathcal{U}^2}$, as the analysis of Montasser et al. (2021) only applies on perturbations up to half the margin; hence, this approach gains no advantage from rejection.

## 5 DEFENSE BY TRANSDUCTION AND REJECTION

The analysis of Theorem 4.1 suggests the following defense algorithm: (1) first obtain a classifier $h$ that is robust and correct on the training data and also robust on the test inputs, (2) then transform $h$ to a selective classifier $\hat{h}$ by rejecting inputs too close to the decision boundary of $h$. We describe the resulting defense below, which we refer to as **TLDR** (Transductive Learning Defense with Rejection).

**Step (1)** To get $h$, we perform adversarial training on both the training set and the test set, using a robust cross-entropy objective. As in TADV (Chen et al., 2022) we train with private randomness. Specifically, we train a model with softmax output as the class prediction probabilities $h^s$ and the class prediction is $h(x) = \arg\max_{y \in \mathcal{Y}} h^s_y(x)$. Given the labeled training data $(\boldsymbol{x}, \boldsymbol{y})$ and the test inputs $\tilde{z}$, we optimize the following objective:

$$\min_h \frac{1}{n} \sum_{(x,y) \in (\boldsymbol{x}, \boldsymbol{y})} \left[ \mathcal{L}_{\text{CE}}(h^s(x), y) + \max_{x' \in \mathcal{U}(x)} \mathcal{L}_{\text{CE}}\left(h^s(x'), y\right) \right] + \frac{\lambda}{m} \sum_{\tilde{z} \in \tilde{\boldsymbol{z}}} \left[ \max_{\tilde{z}' \in \mathcal{U}(\tilde{z})} \mathcal{L}_{\text{CE}}\left(h^s(\tilde{z}'), h(\tilde{z})\right) \right] \quad (1)$$

where $\mathcal{L}_{\text{CE}}$ is the cross-entropy loss and $\lambda > 0$ is a hyper-parameter.

**Step (2)** Having learned $h$, we now turn $h$ into a selective classifier $\hat{h}$. Recall that $\hat{h}$ rejects the input $x$ if there exists $x' \in \mathcal{U}^{1/3}(x)$ with $h(x) \neq h(x')$; otherwise accepts and predicts the label $h(x)$. So we only need to determine the existence of $x' \in \mathcal{U}^{1/3}(x)$ with $h(x) \neq h(x')$. We use a standard inductive attack, PGD, for this by solving:

$$\arg\max_{x' \in \mathcal{U}^{1/3}(x)} \mathcal{L}_{\text{CE}}(h^s(x'), h(x)). \quad (2)$$

When $\mathcal{U}$ is $\ell_p$ norm ball of radius $\epsilon$, the constraint is then $\|x' - x\| \leq \epsilon/3$. In practice, we can generalize this to $\|x' - x\| \leq \epsilon_{\text{defense}}$ where $\epsilon_{\text{defense}}$ is a hyper-parameter we call the *rejection radius*.

### 5.1 ADAPTIVE ATTACKS

Since no strong adaptive attacks exist for the new transduction+rejection setting to our knowledge, we design one here. Our attack is based on GMSA in Chen et al. (2022), which has been shown to be a strong attack for transductive defense (without rejection). The goal of the attack is to find perturbations $\tilde{z}$ of the clean test inputs $\tilde{x}$ such that the transductive learner has a large error when given $(\boldsymbol{x}, \boldsymbol{y}, \tilde{z})$. GMSA runs in stages; in each stage $t$, it simulates the transductive learner on the current data set $(\boldsymbol{x}, \boldsymbol{y}, \tilde{z}_t)$ to get a classifier $h_t$, and then maximizes the minimum or average loss of $\{h_i\}_{i=1}^t$ to get the updated perturbations of the test inputs $\tilde{z}_{t+1}$ (called GMSA$_{\text{MIN}}$ and GMSA$_{\text{AVG}}$, respectively). See Chen et al. (2022) for the details.

GMSA does not directly apply to our setting since we have selective classifiers $\hat{h}$ with a rejection option which is not considered in GMSA. Our contribution is to design a method to get the updated perturbations $\tilde{z}$ of the test inputs in each stage such that the selective classifier incurs a large error. Recall that $\hat{h}$ constructed from $h$ incurs error in two cases: (1) it accepts $\tilde{z}$ and misclassifies with $h(\tilde{z}) \neq y$; (2) $\tilde{z} = \tilde{x}$ and it rejects $\tilde{z}$. We consider the two cases below.

**Case (1)** We will propose a novel loss measuring the loss of a selective classifier $\hat{h}$ on a perturbation $(\tilde{z}, y)$ from a clean test point $(\tilde{x}, y)$ for such kind of error; maximizing this loss gives the desired $\tilde{z}$. Recall that we need $\tilde{z}$ to be accepted and also the prediction $h(\tilde{z}) \neq y$. For the latter, we can

maximize $\mathcal{L}_{\text{CE}}(h^s(\tilde{x}), y)$ where $h^s$ is the class probabilities of $h$ (i.e., its softmax output). The former is equivalent to $\min_{h(\tilde{z}') \neq h(\tilde{z})} \|\tilde{z} - \tilde{z}'\| \geq \epsilon_{\text{defense}}$.

Now, suppose $\mathcal{L}_{\text{DB},h}(\tilde{z}')$ is a *surrogate loss* function on the closeness to the decision boundary; it increases when $\tilde{z}'$ gets closer to the decision boundary of $h$. Then the condition is equivalent to $\|\tilde{z} - p(\tilde{z})\| = \epsilon_{\text{defense}}$ where $p(\tilde{z}) = \arg\max_{\|\tilde{z}' - \tilde{z}\| \leq \epsilon_{\text{defense}}} \mathcal{L}_{\text{DB},h}(\tilde{z}')$. Now, as the maximum value of $\|\tilde{z} - p(\tilde{z})\|$ is exactly $\epsilon_{\text{defense}}$, we would like to maximize $\|\tilde{z} - p(\tilde{z})\|$ to satisfy the condition.

Summing up, for this case, we would like to maximize:

$$\mathcal{L}_{\text{REJ}}(\tilde{z}, y) := \mathcal{L}_{\text{CE}}(h^s(\tilde{z}), y) + \lambda' \|\tilde{z} - p(\tilde{z})\|, \text{ where } p(\tilde{z}) \quad = \quad \arg\max_{\|\tilde{z}' - \tilde{z}\| \leq \epsilon_{\text{defense}}} \mathcal{L}_{\text{DB},h}(\tilde{z}') \tag{3}$$

and $\lambda' > 0$ is a hyper-parameter. Finally, for $\mathcal{L}_{\text{DB},h}$, the following definition works well in our experiments: $\mathcal{L}_{\text{DB},h}(\tilde{z}') := \text{rank}_2 \, h^s(\tilde{z}') - \max h^s(\tilde{z}')$, which is maximized at the decision boundary as the top-two class probabilities are equal.

**Case (2)** A critical step in an effective application of $\mathcal{L}_{\text{REJ}}$ to a transductive attack is the selection of which points to perturb. To do this, we apply a post-processing step after finding $\tilde{z}$ by maximizing (equation 3). We must predict whether $\hat{h}$ is more likely to incur error on $\tilde{z}$ or on the clean input $\tilde{x}$ (i.e., $\hat{h}(\tilde{x}) \neq y$). If we expect that the clean point is likely to be incorrectly classified or rejected, then we update $\tilde{z}$ to $\tilde{x}$. In GMSA, we have access to a series of models trained on previous attack iterations; we estimate the likelihood of success at $\tilde{z}$ and $\tilde{x}$ by the fraction of previous models which fail at each point.

Combining these cases with GMSA gives our final attack (details in Algorithm 1 in Appendix B.5).

# 6 EXPERIMENTS

This section performs experiments to evaluate the proposed method TLDR and compare it with baseline methods (e.g., those using only rejection or transduction). Our main findings are: **1)** TLDR outperforms the baselines significantly in robustness, confirming the advantage of combining transduction and rejection. **2)** Our adaptive attack is significantly stronger than existing attacks which were not designed for the new setting, providing a strong evaluation. **3)** Rejection rates rise steadily with the rejection radius, but few clean samples are rejected and the robust accuracy remains stable.

## 6.1 DATASETS AND DEFENSE/ATTACK SETUP

We evaluate on MNIST (LeCun, 1998) and CIFAR-10 (Krizhevsky et al., 2009). We consider an adversarial budget of $\epsilon = 0.3$ in $l_\infty$ on MNIST and $\epsilon = 8/255$ in $l_\infty$ on CIFAR-10 and CIFAR-100. For defense, on MNIST, we use a LeNet architecture; on CIFAR-10 we use a ResNet-20 architecture. In both cases, we train for 40 epochs with a learning rate of 0.001 using ADAM for optimization. On MNIST, we use 40 iterations of PGD during training with a step size of 0.01. On CIFAR-10, we use 10 iterations of PGD in training with a step size of 2/255. In training TLDR, we set $\lambda = 0.176$ after a warm start period in which $\lambda = 0$. We use a rejection radius of $\epsilon/4$ for selective classifiers. For attack, we use 10 iterations of GMSA on both datasets. On MNIST, we use 200 steps of PGD with a stepsize of 0.01 while generating adversarial examples. On CIFAR-10, the PGD attacks use 100 steps with a stepsize of 1/255. Defense settings used while training models in GMSA (including internal PGD settings) are the standard defense settings. Internal optimizations in the calculation of $\mathcal{L}_{\text{REJ}}$ use 10 steps of PGD with a stepsize of 15% of the rejection radius. We use $\lambda' = 1$ in $\mathcal{L}_{\text{REJ}}$; we observe little sensitivity to the parameter.

## 6.2 ATTACK EVALUATION

Table 3 shows the results of different attack methods on TLDR. Previous work (Chen et al., 2022) shows that transduction-aware attacks are necessary against transductive defenses; we observe that attacks (PGD on $\mathcal{L}_{\text{CE}}$ or $\mathcal{L}_{\text{REJ}}$ and AutoAttack) from the traditional setting perform poorly against our defense. We can also see that GMSA significantly outperforms even a rejection-aware transfer attack (referred to as PGD targeting $\mathcal{L}_{\text{REJ}}$; note that PGD and AutoAttack do *not* target the final model in this case, given the transductive setting, but instead target a proxy trained by the adversary); see Algorithm 2 in Appendix B.5 for the full details.

| Attack | MNIST | CIFAR-10 |
|---|---|---|
| PGD ($\mathcal{L}_{CE}$) | 0.991 | 0.794 |
| PGD ($\mathcal{L}_{REJ}$) | 0.988 | 0.781 |
| AutoAttack | 0.989 | 0.756 |
| GMSA ($\mathcal{L}_{CE}$) | 0.988 | 0.853 |
| **GMSA ($\mathcal{L}_{REJ}$)** | **0.972** | **0.739** |

Table 3: Robust accuracy by different attacks on TLDR. The strongest attack is **boldfaced**.

| Loss | MNIST | CIFAR-10 |
|---|---|---|
| AutoAttack (Croce & Hein, 2020) | 0.980 | 0.592 |
| $\mathcal{L}_{CE}$ | 0.977 | 0.524 |
| $\mathcal{L}_{REJ}(\mathcal{L}_{CE})$ | 0.974 | 0.470 |
| $\mathcal{L}_{REJ}$ | **0.973** | **0.458** |

Table 4: Robust accuracy under different attack losses on a fixed adversarially trained model with rejection, AutoAttack for comparison. The strongest attack is **boldfaced**.

This shows that GMSA is critical for attacking a transductive defender; while PGD and AutoAttack are strong against an inductive model, they performs poorly facing transduction. Finally, we observe that GMSA with $\mathcal{L}_{CE}$ is much weaker than GMSA with $\mathcal{L}_{REJ}$. This shows another key component in our adaptive attack, the loss $\mathcal{L}_{REJ}$, is also critical to get a strong attack against our defense.

To further investigate the importance of $\mathcal{L}_{REJ}$, we attack an adversarially trained model with rejection with PGD on different losses: $\mathcal{L}_{REJ}$, cross-entropy $\mathcal{L}_{CE}$, and $\mathcal{L}_{REJ}$ with $\mathcal{L}_{DB,h}$ replaced by $\mathcal{L}_{CE}$, with AutoAttack given for comparison. Table 4 shows that $\mathcal{L}_{REJ}$ significantly outperforms both PGD targeting alternative losses and AutoAttack. See Appendix C for an evaluation of the effectiveness with which $\mathcal{L}_{REJ}$ targets rejection using the binarization test (Zimmermann et al., 2022).

### 6.3 ROBUSTNESS OF TLDR

| Setting | Defense | Attacker | MNIST | | CIFAR-10 | |
|---|---|---|---|---|---|---|
| | | | $p_{REJ}$ | Robust accuracy | $p_{REJ}$ | Robust accuracy |
| Induction | AT (Madry et al., 2018) | AutoAttack | – | 0.897 | – | 0.448 |
| Rejection only | AT (with rejection) | PGD ($\mathcal{L}_{REJ}$) | 0.852 | 0.968 | 0.384 | 0.634 |
| Transduction only | RMC (Wu et al., 2020) | GMSA ($\mathcal{L}_{CE}$) | – | 0.588 | – | 0.396 |
| | DANN (Ganin et al., 2016) | GMSA ($\mathcal{L}_{CE}$) | – | 0.062 | – | 0.055 |
| | TADV (Chen et al., 2022) | GMSA ($\mathcal{L}_{CE}$) | – | 0.943 | – | 0.541 |
| Transduction+Rejection | URejectron (Goldwasser et al., 2020) | GMSA ($\mathcal{L}_{DISC}$) | 0.274 | 0.721 | **0.000** | 0.145 |
| Transduction+Rejection | **TLDR (ours)** | GMSA ($\mathcal{L}_{REJ}$) | **0.126** | **0.972** | 0.208 | **0.739** |

Table 5: Results on MNIST and CIFAR-10. Robust accuracy is 1 - robust error; see Section 3. $p_{REJ}$ is the percentage of inputs rejected. The baseline results are from (Chen et al., 2022). The strongest attack against each defense is shown. The best result is **boldfaced**.

| Setting | Defense | Architecture | Attacker | CIFAR-10 | | CIFAR-100 | |
|---|---|---|---|---|---|---|---|
| | | | | $p_{REJ}$ | Robust accuracy | $p_{REJ}$ | Robust accuracy |
| Induction | Peng et al. (2023) | Ra WideResNet-28-10 | AutoAttack | – | 0.651 | – | 0.372 |
| Induction | Peng et al. (2023) | Ra WideResNet-70-16 | AutoAttack | – | 0.711 | – | 0.388 |
| Induction | Wang et al. (2023) | WideResNet-28-10 | AutoAttack | – | 0.673 | – | 0.388 |
| Induction | Wang et al. (2023) | WideResNet-70-16 | AutoAttack | – | 0.707 | – | 0.427 |
| Transduction+Rejection | **TLDR (ours)** | ResNet-20 | GMSA ($\mathcal{L}_{REJ}$) | 0.208 | 0.739 | – | – |
| Transduction+Rejection | **TLDR (ours)** | WideResNet-28-10 | GMSA ($\mathcal{L}_{REJ}$) | **0.111** | **0.816** | 0.171 | **0.579** |

Table 6: Comparison with state-of-the-art (Croce et al., 2020). The best result is **boldfaced**.

**Baselines.** (1) AT: adversarial training (Madry et al., 2018); (2) AT (with rejection): adversarial training (AT) with rejection; (3) RMC (Wu et al., 2020); (4) DANN (Ganin et al., 2016); (5) TADV (Chen et al., 2022); (6) Rejectron (Goldwasser et al., 2020). Among them, (1) is in the traditional induction setting, (2) is rejection only, (3)(4)(5) are transduction only, and (6) incorporates both transduction and rejection.

**Evaluation.** We attack the defenses and report the robust accuracy (1 - the robust error defined in Section 3). To attack inductive classifiers, we use AutoAttack (Croce & Hein, 2020). For inductive selective classifiers, we use PGD on the rejection-aware loss $\mathcal{L}_{REJ}$ from Eqn (3). For transductive

| TLDR Components | | Attacker | MNIST | | CIFAR-10 | |
|---|---|---|---|---|---|---|
| Rejection | $L_{\text{test}}$ | | $p_{\text{REJ}}$ | Robust accuracy | $p_{\text{REJ}}$ | Robust accuracy |
| ✓ | ✓ | GMSA ($\mathcal{L}_{\text{REJ}}$) | **0.588** | 0.967 | 0.208 | **0.739** |
| ✓ | ✗ | GMSA ($\mathcal{L}_{\text{REJ}}$) | 0.646 | **0.975** | **0.179** | 0.725 |
| ✗ | ✓ | GMSA ($\mathcal{L}_{\text{CE}}$) | – | 0.900 | – | 0.516 |
| ✗ | ✗ | GMSA ($\mathcal{L}_{\text{CE}}$) | – | 0.935 | – | 0.516 |

Table 7: Ablation study of TLDR. The best result is **boldfaced**.

classifiers, we use GMSA which has been shown to be a strong adaptive attack on transduction (Chen et al., 2022). Finally, for our transductive selective classifiers, we use our adaptive attack in Section 5.1 (roughly GMSA with $\mathcal{L}_{\text{REJ}}$). For Rejectron (Goldwasser et al., 2020) we use GMSA with a loss function $\mathcal{L}_{\text{DISC}}$ targeting their defense; see Appendix B.6 for the details. For transductive models, we report the stronger of GMSA$_{\text{MIN}}$ and GMSA$_{\text{AVG}}$. Inductive models are trained with standard adversarial training (Goodfellow et al., 2015), and transductive models with the TLDR loss in Eqn (1). As Rejectron depends heavily on a key hyperparameter determining confidence needed to reject, we report the results for the parameter value strongest against our attack. The best-performing value on CIFAR-10 effectively eliminated the possibility of rejection (hence the rejection rate of 0); other choices resulted in near-0 robust accuracy.

**Comparison of Defenses.** Table 5 shows the robust accuracy and rejection rate of different methods. We observe that either transduction or rejection can improve the performance, while combining both techniques leads to the best results. In particular, our defense outperforms existing transductive defenses such as RMC and DANN. Results for $l_2$ perturbations are given for $l_2$ in Appendix C. See Table 6 for a comparison to the state-of-the-art. With a much smaller ResNet-20 architecture, TLDR outperforms the strongest existing baseline on CIFAR-10; with a WideResNet-28-10 architecture we obtain improvements in robust accuracy above the state-of-the-art of over 10% and 15%, respectively, on CIFAR-10 and CIFAR-100.

**Discussion on Evaluation.** As our key focus is on demonstrating the potential advantages of one setting (transduction+rejection) over others, comparisons between settings are necessary. In each setting, robust accuracy represents the same concept, the fraction of samples on which we are correct. The difference between settings lies in their different notions of "correctness"; each concept of correctness incorporates both the potential advantages and the disadvantages of each setting, e.g. in the rejection case, a new type of error is possible: rejecting a clean sample. Hence, we compare the fraction of samples on which we can be correct between settings (and between defenses in the same setting).

### 6.4 ABLATION STUDY OF TLDR

Compared to traditional defenses, TLDR has two novel components: using the given test inputs in training the classifier (the second term in Equation (1), referred to as $L_{\text{test}}$), and transforming the trained classifier into one with rejection. Table 7 shows the results of the ablation study on these two components. In all cases, rejection significantly improves results. The use of transduction is helpful on CIFAR-10, but reduces performance on MNIST, potentially due to the lower difficulty of deriving robust predictions on MNIST; hence, the knowledge of the specific test inputs is less useful.

## 7 CONCLUSION

Existing works on leveraging transduction and rejection gave mixed results on their benefits for adversarial robustness. In this work we take a step in realizing their promise in practical deep learning settings. Theoretically, we show that a novel application of Tramèr's results give improved *sample complexity* for robust learning in the bounded perturbations setting. Guided by our theory, we identified a practical robust learning algorithm leveraging both transduction and rejection. Systematic experiments confirm the benefits of our constructions. There are many future avenues to explore, such as improving the theoretical bounds, and improving the efficiency of our algorithms.

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

# Appendix

## A    PROOF DETAILS

Before introducing the proof for the generalization results, we first need to make some additional definitions. We define the *empirical robust risk* as

$$\hat{R}_{\mathcal{U}}(h; S) = \sum_{(x,y) \in S} \left[ \sup_{z \in \mathcal{U}(x)} \mathbb{1}\{h(z) \neq y\} \right]$$

And we can define the *empirical robust risk under rejection* accordingly:

$$\hat{R}_{\mathcal{U}}^{\text{rej}}(h; S) = \sum_{(x,y) \in S} \left[ \sup_{z \in \mathcal{U}(x)} \mathbb{1}\{h(x) \neq y \vee h(z) \notin \{y, \perp\}\} \right]$$

And we can define the corresponding robust empirical risk minimization procedure (under rejection) as follows:

$$\text{RERM}_{\mathcal{H}}(S) := \underset{h \in \mathcal{H}}{\text{argmin}} \, \hat{R}_{\mathcal{U}}(h; S)$$

$$\text{RERM}_{\mathcal{H}}^{\text{rej}}(S) := \underset{h \in \mathcal{H}}{\text{argmin}} \, \hat{R}_{\mathcal{U}}^{\text{rej}}(h; S)$$

### A.1    REJECTION ONLY: REALIZABLE CASE

**Definition A.1** (Realizable Robust PAC Learnability under Rejection)**.** For $\mathcal{Y} = \{0, 1\}$, $\forall \epsilon, \delta \in (0, 1)$, $\mathcal{H} = \mathcal{H}_c \times \mathcal{H}_r$, the sample complexity of realizable robust $(\epsilon, \delta)$ - PAC learning of $\mathcal{H}$ with respect adversary $\mathcal{U}$ under rejection, denoted as $\mathcal{M}_{\text{RE}}(\epsilon, \delta; \mathcal{H}, \mathcal{U})$, is defined as the smallest $m \in \mathbb{N} \cup \{0\}$ for which there exists a learning rule $\mathcal{A} : (\mathcal{X} \times \mathcal{Y})^m \longmapsto (\mathcal{Y} \cup \{\perp\})^{\mathcal{X}}$ s.t. for every data distribution $\mathcal{D}$ over $(\mathcal{X} \times \mathcal{Y})^m$ where there exists a predictor with rejection option $h^* \in \mathcal{H}$ with 0 risk, $R_{\mathcal{U}, \text{rej}}(h^*; \mathcal{D}) = 0$ with probability at least $1 - \delta$ over $S \sim \mathcal{D}^m$,

$$R_{\mathcal{U}}^{\text{rej}}(\mathcal{A}(S); \mathcal{D}) \leq \epsilon$$

If no such $m$ exists, $\mathcal{M}_{\text{RE}}(\epsilon, \delta; \mathcal{H}, \mathcal{U}) = \infty$. We say that $\mathcal{H}$ is robustly PAC learnable under rejection in the realizable setting with respect to adversary $\mathcal{U}$ if $\forall \epsilon, \delta \in (0, 1)$, $\mathcal{M}_{\text{RE}}(\epsilon, \delta; \mathcal{H}, \mathcal{U})$ is finite.

**Theorem A.2** (Sample Complexity for Realizable Robust PAC Learning under Rejection)**.** *In the realizable setting, for any $\mathcal{H} = \mathcal{H}_c \times \mathcal{H}_r$ and $\mathcal{U}$, and any $\epsilon, \delta \in (0, 1/2)$,*

$$\mathcal{M}_{RE}(\epsilon, \delta; \mathcal{H}, \mathcal{U}) = 2^{O\left((d_r + d_c) \log(d_r + d_c)\right)} \frac{1}{\varepsilon} \log\left(\frac{1}{\varepsilon}\right) + O\left(\frac{1}{\varepsilon} \log\left(\frac{1}{\delta}\right)\right) \tag{4}$$

*where $d_r = \text{VC}(\mathcal{H}_r), d_c = \text{VC}(\mathcal{H}_c)$.*

*The idea of the proof is to adapt the classical sample compression argument (Littlestone & Warmuth, 1986) with improvements based on (Montasser et al., 2019; Hanneke et al., 2019; Moran & Yehudayoff, 2016). The generalization result in the inductive case directly comes from Equation (29).*

*Proof.* First, we define the concept of *sample compression scheme* and *sample compression algorithm*.

**Definition A.3** (Sample Compression Scheme)**.** Given $\forall m \in \mathbb{N}$ samples, $S \sim \mathcal{D}^m$, a *sample compression scheme* of size $k$ is defined by the following pair of functions:

1. Compression function $\kappa : (\mathcal{X} \times \mathcal{Y})^m \mapsto (\mathcal{X} \times \mathcal{Y})^{\leq k}$.

2. Reconstruction function: $\rho : (\mathcal{X} \times \mathcal{Y})^{\leq k} \mapsto \mathcal{H}$.

An algorithm $\mathcal{A}$ is a *sample compression algorithm* if $\exists \kappa, \rho$ s.t. $\mathcal{A}(S) = (\kappa \circ \rho)(S)$.

Fix $\epsilon, \delta \in (0, 1)$, $m > 2(d_r + d_c) \log(d_r + d_c)$. Let the compression parameter, $n = O\left((d_r + d_c) \log(d_r + d_c)\right)$. Let $\mathcal{D}$ be any distribution, then by realizability of the learner, $\inf_{h \in \mathcal{H}} R_{\mathcal{U}}^{\text{rej}}(h; \mathcal{D}) = 0$. Thus, $\forall S$ sampled from $\mathcal{D}$, we have $\hat{R}_{\mathcal{U}}^{\text{rej}}(\text{RERM}_{\mathcal{H}}^{\text{rej}}(S); S) = 0$.

**Compression** First, we define a compression function $\kappa$ as through the following inflation and discretization procedure. Given the training data $S := \{(x_i, y_i)\}_{i \in [m]}$, we define the following index mapping:

$$I(x) = \min\{i \in [m] : x \in \mathcal{U}(x_i)\}, \quad \forall x \in \bigcup_{i \in [m]} \mathcal{U}(x_i). \tag{5}$$

In another word, this index function outputs the first indexed training sample to include $x$ in its neighborhood.

Then, we consider the set of RERM mapping learned by a size $n$ subset of the training data:

$$\hat{\mathcal{H}} = \{\text{RERM}_{\mathcal{H}}^{\text{rej}}(L) : L \subseteq S, |L| = n\}. \tag{6}$$

Note that

$$|\hat{\mathcal{H}}| \leq |\{L : L \subseteq S, |L| = n\}| = \binom{m}{n} \leq \left(\frac{em}{n}\right)^n. \tag{7}$$

Then, we inflate the data in the following way:

$$S_{\mathcal{U}} = \bigcup_{i \in [m]} \{(x_{I(x)}, x, y_{I(x)}) : x \in \mathcal{U}(x_i)\}. \tag{8}$$

Note that $x_{I(x)}$ can be different from $x_i$.

Let's define the following transformation $T$:

$$T(h)(x, x', y) := \mathbb{1}\{h(x) \neq y \vee h(x') \notin \{y, \perp\}\}, \ h \in \mathcal{H}. \tag{9}$$

And we can obtain the transformed hypothesis class $T(\mathcal{H}) := \{T(h)|h \in \mathcal{H}\}$.

Now, we proceed to define the *dual space* $\mathcal{G}$ of $T(\mathcal{H})$ as the following set of functions.

$$\mathcal{G} := \{g_{(x,x',y)}|g_{(x,x',y)}(t) = t(x, x', y), \ t \in T(\mathcal{H})\}. \tag{10}$$

We denote the VC dimension of the dual space as $\text{VC}^*(T(\mathcal{H})) := \text{VC}(\mathcal{G})$.

By Lemma A.1,

$$\text{VC}(T(\mathcal{H})) = O\left((d_r + d_c)\log(d_r + d_c)\right). \tag{11}$$

By the classic result in (Assouad, 1983), the VC dimension of the dual space satisfies the following inequality:

$$VC^*(T(\mathcal{H})) < 2^{\text{VC}(T(\mathcal{H}))+1}. \tag{12}$$

Now, we can construct the compressed dataset $\hat{S}_{\mathcal{U}}$ as the following. For each $(x, x', y) \in S_{\mathcal{U}}$, $\{g_{(x,x',y)}(t)\}_{t \in T(\hat{\mathcal{H}})}$ gives a labeling. When ranging over $(x, x', y) \in S_{\mathcal{U}}$, the labeling may not be unique. So for each unique labeling, we choose a representative $(x, x', y) \in S_{\mathcal{U}}$, and let $\hat{S}_{\mathcal{U}}$ be the set of the representatives. That is:

$$\hat{S}_{\mathcal{U}} = \left\{(x, x', y) \in S_{\mathcal{U}} \ \middle| \ \{g_{(x,x',y)}(t)\}_{t \in T(\hat{\mathcal{H}})} \text{ provides a unique labeling}\right\}. \tag{13}$$

Intuitively, $\hat{S}_{\mathcal{U}}$ split the infinite size dataset $S_{\mathcal{U}}$ into finite size according to the labeling of $T(\hat{\mathcal{U}})$ on the dual space. Thus, $\hat{S}_{\mathcal{U}}$ is not necessarily unique but always exists. And $|\hat{S}_{\mathcal{U}}|$ equals the number of possible labeling for $T(\hat{\mathcal{H}})$.

Let $d_* := VC(\mathcal{G}) = VC^*(T(\mathcal{H}))$ denote the VC-dimension of $\mathcal{G}$, the dual hypothesis class of $T(\hat{\mathcal{H}})$ (Assouad, 1983). By applying Sauer's Lemma, we obtain that for $|T(\hat{\mathcal{H}})| > d_*$,

$$|\hat{S}_{\mathcal{U}}| \leq \left(\frac{e|T(\hat{\mathcal{H}})|}{d_*}\right)^{d_*}. \tag{14}$$

Let $n = \Theta\left(\text{VC}\left(T\left(\mathcal{H}\right)\right)\right)$. For $m \geq n$, we have

$$|\hat{S}_{\mathcal{U}}| \leq \left(e|T(\hat{\mathcal{H}})|\right)^{d_*} \tag{15}$$

$$\leq \left(e|\hat{\mathcal{H}}|\right)^{d_*} \tag{16}$$

$$\leq \left(e\left(\frac{em}{n}\right)^n\right)^{d_*} \tag{17}$$

$$\leq \left(\frac{e^2 m}{n}\right)^{nd_*} \tag{18}$$

$$= \left(\frac{e^2 m}{\text{VC}(T(\mathcal{H}))}\right)^{\Theta(\text{VC}(T(\mathcal{H})) \cdot \text{VC}(T(\mathcal{H}^*)))}. \tag{19}$$

Now we have obtain the compression map: $\kappa(S) = \hat{S}_{\mathcal{U}}$.

**Reconstruction**   Now, we want to reconstruct a hypothesis from $\hat{S}_{\mathcal{U}}$. First, suppose we have a data distribution over $\hat{S}_{\mathcal{U}}$, denoted as $\mathcal{P}$. This distribution $\mathcal{P}$ over samples will be later used in the $\alpha-$boosting procedure.

Then, we sample the set of $n$ i.i.d. samples from $\mathcal{P}$ and obtain $S' \in \hat{S}_{\mathcal{U}}$. By classic PAC learning guarantee (Blumer et al., 1989), for $n = \Theta(\text{VC}(T(\mathcal{H}))) = \Theta(d_r + d_c) \log(d_r + d_c)$, we have with non-zero probability $\forall t \in T(\mathcal{H})$ with $\sum_{(x,x',y) \in S'} t(x, x', y) = 0$ implies $\mathbb{E}_{(x,x',y) \sim \mathcal{P}} t(x, x', y) < 1/9$. Let $L = \{(x, y) : (x, x', y) \in S'\} \subseteq S$, and $t_{\mathcal{P}} = T(\text{RERM}_{\mathcal{H}}^{\text{rej}}(L))$. Since $\hat{R}_{\mathcal{U}}^{\text{rej}}(\text{RERM}_{\mathcal{H}}^{\text{rej}}(L); L) = 0$, $\forall (x, x', y) \in S', t_{\mathcal{P}}(x, x', y) = 0$. Thus, $\forall \mathcal{P}$ over $\hat{S}_{\mathcal{U}}$, there exists a weak learner $t_{\mathcal{P}} \in T(\hat{\mathcal{H}})$, s.t. $\mathbb{E}_{(x,x',y) \sim \mathcal{P}} t_{\mathcal{P}}(x, x', y) < 1/9$.

Now, we use $t_{\mathcal{P}}$ as a *weak hypothesis* in a boosting algorithm, specifically $\alpha-$boost algorithm from (Schapire & Freund, 2012) with $\hat{S}_{\mathcal{U}}$ as the dataset and $\mathcal{P}_k$ generated at each round of the algorithm. Then with appropriate choice of $\alpha$, running $\alpha-$boosting for $K = O(\log(|\hat{S}_{\mathcal{U}}|))$ rounds gives a sequence of hypothesis $h_1, \ldots, h_K \in \hat{\mathcal{H}}$ and the corresponding $t_i = T(h_i)$ such that $\forall (x, x', y) \in \hat{S}_{\mathcal{U}}$,

$$\frac{1}{K} \sum_{k=1}^{K} \mathbb{1}\{h_k(x) \neq y \vee h_k(x') \notin \{y, \bot\}\} \tag{20}$$

$$= \frac{1}{K} \sum_{k=1}^{K} t_k(x, x', y) \tag{21}$$

$$< \frac{2}{9} < \frac{1}{3}. \tag{22}$$

Since $\hat{S}_{\mathcal{U}}$ includes all the unique labellings, $\frac{1}{K} \sum_{k=1}^{K} t_k(x, x', y) < \frac{1}{3}$, $\forall (x, x', y) \in \hat{S}_{\mathcal{U}}$ implies

$$\frac{1}{K} \sum_{k=1}^{K} t_k(x, x', y) < \frac{1}{3}, \ \forall (x, x', y) \in S_{\mathcal{U}}. \tag{23}$$

Let $\bar{h} := \text{Majority}(h_1, \ldots, h_K)$, i.e., $\bar{h}$ outputs the prediction in $\mathcal{Y} \cup \{\bot\}$ that receives the most votes from $\{h_1, \ldots, h_K\}$. Then $\forall (x, x', y) \in \hat{S}_{\mathcal{U}}$,

$$\mathbb{1}\{\bar{h}(x) \neq y \vee \bar{h}(x') \notin \{y, \bot\}\} = 0. \tag{24}$$

This is because: (1) on $x$, less than $1/3$ of $h_i$'s do not output $y$, so $\bar{h}(x) = y$; (2) on $x'$, less than $1/3$ of $h_i$'s do not output $y$ or $\bot$, so the majority vote must be in $y$ or $\bot$, i.e., $\bar{h}(x) \in \{y, \bot\}$.

In summary, given the same $m$ training samples, we can simply find a $\bar{h}$ with 0 robust error on $S$:

$$\hat{R}_{\mathcal{U}}^{\text{rej}}(\bar{h}; \mathcal{D}) = \sum_{i=1}^{m} \left[ \sup_{z \in \mathcal{U}(x)} \mathbb{1}\{\bar{h}(x) \neq y \vee \bar{h}(z) \notin \{y, \bot\}\} \right] = 0. \tag{25}$$

Now we have the compression set with size:

$$nK = O(\text{VC}(T(\mathcal{H})) \log(|\hat{S}_{\mathcal{U}}|)) = O(\text{VC}(T(\mathcal{H}))^2 \text{ VC}^*(T(\mathcal{H})) \log(m/\text{VC}(T(\mathcal{H}))))$$

Then, we apply Lemma 11 of (Montasser et al., 2019) (Replacing $R_{\mathcal{U}}$ with $R_{\mathcal{U}}^{\text{rej}}$ still holds), we obtain for sufficiently large $m$, with probability at least $1 - \delta$,

$$R_{\mathcal{U}}^{\text{rej}}(\bar{h}; \mathcal{D}) \le O\left(\text{VC}(T(\mathcal{H}))^2 \text{ VC}^*(T(\mathcal{H})) \frac{1}{m} \log(m/\text{VC}(T(\mathcal{H}))) \log(m) + \frac{1}{m} \log(1/\delta)\right). \qquad (26)$$

We then can extend the sparsification procedure from (Moran & Yehudayoff, 2016; Montasser et al., 2019) to the rejection scenario. Since $t_1, \ldots, t_K \in T(\hat{\mathcal{H}})$, the classic uniform convergence results (Shalev-Shwartz & Ben-David, 2014) implies that we can sample $N = O(\text{VC}^*(T(\mathcal{H})))$ i.i.d. indices $i_1, \ldots, i_N \sim \text{Uniform}([K])$ and obtain:

$$\sup_{(x,x',y) \in S_{\mathcal{U}}} \left| \frac{1}{N} \sum_{j=1}^{N} t_{i_j}(x, x', y) - \frac{1}{K} \sum_{i=1}^{T} t_i(x, x', y) \right| < \frac{1}{18} \qquad (27)$$

And thus, we can combine Equation (20) with Equation (27) and obtain:

$$\forall (x, x', y) \in S_{\mathcal{U}}, \frac{1}{N} \sum_{j=1}^{N} t_{i_j}(x, x', y) \le -\frac{1}{18} + \frac{1}{K} \sum_{i=1}^{K} t_k(x, x', y) < -\frac{1}{18} + \frac{4}{9} = \frac{1}{2}$$

we can further obtain an improved hypothesis $\bar{t}' := \text{Majority}(t_{i_1}, \ldots t_{i_N})$ with

$$\bar{t}'(x, x', y) = 0, \forall (x, x', y) \in S_{\mathcal{U}}$$

Thus, the compression set has a reduced size:

$$nN = O(\text{VC}(T(\mathcal{H})) \cdot \text{VC}^*(T(\mathcal{H})))$$

Now, we apply Lemma 11 of (Montasser et al., 2019) and can obtain the following improved bound. Applying similar strategy from Equation (24), we can obtain

$$\bar{h}' := \text{Majority}(h_{i_1}, \ldots h_{i_N}) = \rho(\hat{S}_{\mathcal{U}}) = \mathcal{A}(S) \qquad (28)$$

which is our full reconstruction map.

Then, for large sample size $m \ge c \text{ VC}(T(\mathcal{H})) \text{ VC}^*(T(\mathcal{H}))$ ($c$ is a sufficiently large constant), with probability at least $1 - \delta$,

$$R_{\mathcal{U}, \text{rej}}(\bar{h}'; \mathcal{D}) \le O\left(\text{VC}(T(\mathcal{H})) \text{ VC}^*(\mathcal{H}) \frac{1}{m} \log(m) + \frac{1}{m} \log(1/\delta)\right) \qquad (29)$$

Plugging in Lemma Appendix A.1 and solving for $m$ gives

$$\mathcal{M}_{\text{RE}}(\epsilon, \delta; \mathcal{H}, \mathcal{U}) = 2^{O(\text{VC}(T(\mathcal{H})))} \frac{1}{\varepsilon} \log\left(\frac{1}{\varepsilon}\right) + O\left(\frac{1}{\varepsilon} \log\left(\frac{1}{\delta}\right)\right) \qquad (30)$$

$$= 2^{O((d_r + d_c) \log(d_r + d_c))} \frac{1}{\varepsilon} \log\left(\frac{1}{\varepsilon}\right) + O\left(\frac{1}{\varepsilon} \log\left(\frac{1}{\delta}\right)\right) \qquad (31)$$

$\square$

**Lemma** [VC dimension of robust loss with rejection] Let $\text{VC}(\mathcal{H}_c) = d_c$, and $\text{VC}(\mathcal{H}_r) = d_r$. Then, $\text{VC}(T(\mathcal{H})) = O((d_r + d_c) \log(d_r + d_c))$.

*Proof.* Suppose $d > d_r + d_c$.

By definition of VC dimension, the max number of labeling of $d$ points is $2^d$ on $h \in T(\mathcal{H})$. And since the label of $h$ is a deterministic function of $h_c$ and $h_r$, by Sauer's Lemma, the number of labeling of $h$ is at most $O(d^{d_r}) \times O(d^{d_c}) = O(d^{d_r + d_c})$.

Thus, $2^d = O(d^{d_r + d_c})$. And $d = O((d_r + d_c) \log(d_r + d_c))$.

If $d < d_r + d_c$, $d = O(d_r + d_c) \log(d_r + d_c)$ by definition.

$\square$

## A.2  REJECTION ONLY: AGNOSTIC CASE

Now, we define notion of PAC learnability in the agnostic case under rejection setting as the follows:

**Definition A.4** (Robust PAC Learnability under Rejection). For $\mathcal{Y} = \{0, 1\}, \forall \epsilon, \delta \in (0, 1), \mathcal{H} = \mathcal{H}_c \times \mathcal{H}_r$, the sample complexity of robust $(\epsilon, \delta)$ - PAC learning of $\mathcal{H}$ with respect to perturbation $\mathcal{U}$ under rejection, denoted as $\mathcal{M}_{\mathrm{AG}}(\epsilon, \delta; \mathcal{H}, \mathcal{U})$, is defined as the smallest $m \in \mathbb{N} \cup \{0\}$ for which there exists a learning rule $\mathcal{A} : (\mathcal{X} \times \mathcal{Y})^m \longmapsto (\mathcal{Y} \cup \{\bot\})^{\mathcal{X}}$ s.t. for every data distribution $\mathcal{D}$ over $(\mathcal{X} \times \mathcal{Y})^m$,

$$\mathrm{R}^{\mathrm{rej}}_{\mathcal{U}}(\mathcal{A}(S); \mathcal{D}) \leq \mathrm{OPT}^{\mathrm{rej}}_{\mathcal{U}} + \epsilon$$

with probability at least $1 - \delta$ over $S \sim \mathcal{D}^m$. If no such $m$ exists, $\mathcal{M}_{\mathrm{AG}}(\epsilon, \delta; \mathcal{H}, \mathcal{U}) = \infty$. We say that $\mathcal{H}$ is robustly PAC learnable under rejection if $\mathcal{M}_{\mathrm{AG}}(\epsilon, \delta; \mathcal{H}, \mathcal{U})$ is finite for all $\epsilon, \delta \in (0, 1)$.

**Lemma A.5.** *Let* $\mathcal{M}_{\mathrm{RE}} = \mathcal{M}_{\mathrm{RE}}(1/3, 1/3; \mathcal{H}, \mathcal{U})$. *Then,*

$$\mathcal{M}_{\mathrm{AG}}(\epsilon, \delta; \mathcal{H}, \mathcal{U}) = O\left(\frac{\mathcal{M}_{\mathrm{RE}}}{\epsilon^2} \log^2\left(\frac{\mathcal{M}_{\mathrm{RE}}}{\epsilon}\right) + \frac{1}{\epsilon^2} \log\left(\frac{1}{\delta}\right)\right) \tag{32}$$

*Proof.* The proof detail follows exactly the same from the Proof of Theorem 8 from (Montasser et al., 2019) with the loss replaced. ☐

**Theorem A.6** (Sample Complexity for Agnostic Robust PAC Learning under Rejection). *In the agnostic setting, for any* $\mathcal{H} = \mathcal{H}_c \times \mathcal{H}_r$ *and* $\mathcal{U}$, *and any* $\epsilon, \delta \in (0, 1/2)$,

$$\mathcal{M}_{\mathrm{AG}}(\epsilon, \delta; \mathcal{H}, \mathcal{U}) = O\Big(\mathrm{VC}(T(\mathcal{H}))\,\mathrm{VC}^*(T(\mathcal{H}))\log\left(\mathrm{VC}(T(\mathcal{H}))\,\mathrm{VC}^*(T(\mathcal{H}))\right) \tag{33}$$

$$\frac{1}{\varepsilon^2}\log^2\left(\frac{\mathrm{VC}(T(\mathcal{H}))\,\mathrm{VC}^*(T(\mathcal{H}))}{\varepsilon}\right) + \frac{1}{\varepsilon^2}\log\left(\frac{1}{\delta}\right)\Big) \tag{34}$$

$$= 2^{O(\mathrm{VC}(\mathcal{H}))}\frac{1}{\varepsilon^2}\log^2\left(\frac{1}{\varepsilon}\right) + O\left(\frac{1}{\varepsilon^2}\log\left(\frac{1}{\delta}\right)\right) \tag{35}$$

$$= 2^{O\left((d_r+d_c)\log(d_r+d_c)\right)}\frac{1}{\varepsilon^2}\log^2\left(\frac{1}{\varepsilon}\right) + O\left(\frac{1}{\varepsilon^2}\log\left(\frac{1}{\delta}\right)\right) \tag{36}$$

*where* $d_r = \mathrm{VC}(\mathcal{H}_r), d_c = \mathrm{VC}(\mathcal{H}_c)$.

*Proof.* Combining results from Lemma Lemma A.5 and Theorem A.2 gives the complexity result.

Solving Equation (35) gives the following generalization result given in Table 1

$$\Pr_{(\boldsymbol{x}, \boldsymbol{y}) \sim \mathcal{D}^n}\left[\mathrm{R}^{\mathrm{rej}}_{\mathcal{U}}(\mathcal{A}(\boldsymbol{x}, \boldsymbol{y}); \mathcal{D}) \leq \epsilon\right] \geq 1 - \delta$$

where $\epsilon = O\left(\sqrt{\frac{2^{\mathrm{VC}(T(\mathcal{H}))} + \log(1/\delta)}{n}}\right)$. ☐

## A.3  TRANSDUCTION+REJECTION: REALIZABLE CASE

We will prove a more general result which then implies Theorem 4.1. First, the training data can also be perturbed, i.e., the adversary perturbs $\boldsymbol{z} \in \mathcal{U}(\boldsymbol{x})$ and $\tilde{\boldsymbol{z}} \in \mathcal{U}(\tilde{\boldsymbol{x}})$, and the learner $\mathbb{A}$ are given $(\boldsymbol{z}, \boldsymbol{y}, \tilde{\boldsymbol{z}})$ instead of $(\boldsymbol{x}, \boldsymbol{y}, \tilde{\boldsymbol{x}})$. The criterion in the transductive rejection error (see Table 2) is then the worst case over both $\boldsymbol{z} \in \mathcal{U}(\boldsymbol{x})$ and $\tilde{\boldsymbol{z}} \in \mathcal{U}(\tilde{\boldsymbol{x}})$. Second, we will consider $\mathrm{OPT}_{\mathcal{U}^3} = 0$ and prove the guarantee tolerating $\mathcal{U}^2$. This then implies the guarantee tolerating $\mathcal{U}$ when $\mathrm{OPT}_{\mathcal{U}^{3/2}} = 0$.

In general the set of optimally learned classifiers $\Delta$ is defined as follows Montasser et al. (2021):

$$\Delta^{\mathcal{U}}_{\mathcal{H}}(\boldsymbol{z}, \boldsymbol{y}, \tilde{\boldsymbol{z}}) = \begin{cases} \{h \in \mathcal{H} : \mathrm{R}_{\mathcal{U}^{-1}}(h; \boldsymbol{z}, \boldsymbol{y}) = 0 \wedge \mathrm{R}_{\mathcal{U}^{-1}}(h; \tilde{\boldsymbol{z}}) = 0\} & \text{(Realizable Case)} \\ \underset{h \in \mathcal{H}}{\arg\min}\max\{\mathrm{R}_{\mathcal{U}^{-1}}(h; \boldsymbol{z}, \boldsymbol{y}), \mathrm{R}_{\mathcal{U}^{-1}}(h; \tilde{\boldsymbol{z}})\} & \text{(Agnostic Case)} \end{cases}$$

where

$$\mathrm{R}_{\mathcal{U}}(h; \boldsymbol{z}, \boldsymbol{y}) = \sup_{\tilde{\boldsymbol{x}} \in \mathcal{U}(\boldsymbol{z})}\frac{1}{n}\sum_{i=1}^{n}\mathbb{1}\{h(\tilde{x}_i) \neq y_i\}$$

and

$$R_{\mathcal{U}}(h; z) = R_{\mathcal{U}}(h; z, h(z)).$$

Recall the transformation $F$ which we define following Tramèr Tramer (2022) in Section 4.

Then, we define the *relaxed robust shattering dimension* following Montasser et al. (2021):

**Definition A.7** (Relaxed Robust Shattering Dimension). A sequence $z_1, \ldots, z_k \in \mathcal{X}$ is *relaxed $\mathcal{U}$-robustly shattered* by $\mathcal{H}$, if $\forall y_1, \ldots, y_k \in \{\pm 1\}$: $\exists x_1^{y_1}, \ldots, x_k^{y_k} \in \mathcal{X}$ and $\exists h \in \mathcal{H}$ such that $z_i \in \mathcal{U}(x_i^{y_i})$ and $h(\mathcal{U}(x_i^{y_i})) = y_i$, $\forall 1 \leq i \leq k$. The *relaxed $\mathcal{U}$-robust shattering dimension* $\mathrm{rdim}_{\mathcal{U}}(\mathcal{H})$ is defined as the largest $k$ for which there exist $k$ points that are relaxed $\mathcal{U}$-robustly shattered by $\mathcal{H}$.

Define the set of *intermediate perturbations* as follows:

**Definition A.8** (Intermediate Perturbations). Given $x$ and $z$ and perturbations $\mathcal{U}_1$ and $\mathcal{U}_2$, the set of possible intermediate perturbations between $x$ and $z$ is

$$\mathrm{ip}_{\mathcal{U}_1, \mathcal{U}_2}(x, z) = \begin{cases} \{x\} & \text{if } x = z \\ \mathcal{U}_1(x) \cap \mathcal{U}_2^{-1}(z) & \text{otherwise} \end{cases}$$

**Theorem A.9.** *For any $n \in \mathbb{N}$, $\delta > 0$, class $\mathcal{H}$, perturbation set $\mathcal{U}$, and distribution $\mathcal{D}$ over $\mathcal{X} \times \mathcal{Y}$ satisfying $\mathrm{OPT}_{\mathcal{U}^{-1}\mathcal{U}} = 0$:*

$$\Pr_{\substack{(x,y) \sim \mathcal{D}^n \\ (\tilde{x}, \tilde{y}) \sim \mathcal{D}^n}} \left[ \begin{array}{l} \forall z \in \mathcal{U}^3(x), \forall z_0 \in \mathrm{ip}_{\mathcal{U}, \mathcal{U}^2}(x, z), \forall \tilde{z} \in \mathcal{U}^3(\tilde{x}), \forall \tilde{z}_0 \in \mathrm{ip}_{\mathcal{U}, \mathcal{U}^2}(\tilde{x}, \tilde{z}), \\ \forall \hat{h} \in \mathrm{F}_{\mathcal{U}}\left(\Delta_{\mathcal{H}}^{\mathcal{U}}(z_0, y, \tilde{z}_0)\right) : \mathrm{err}^{\mathrm{rej}}(\hat{h}; x, y, \tilde{x}, \tilde{z}, \tilde{y}) \leq \epsilon \end{array} \right] \geq 1 - \delta$$

*where $\epsilon = \frac{\mathrm{rdim}_{\mathcal{U}^{-1}}(\mathcal{H}) \log(2n) + \log(1/\delta)}{n} \leq \frac{\mathrm{VC}(\mathcal{H}) \log(2n) + \log(1/\delta)}{n}$.*

*Proof.* We adapt the strategy of Theorem 5 of Tramer (2022) for the rejection scenario.

By setting $z = z_0, \tilde{z} = \tilde{z}_0$ and applying Theorem 1 of Montasser et al. (2021), we obtain the following

$$\Pr_{\substack{(x,y) \sim \mathcal{D}^n \\ (\tilde{x}, \tilde{y}) \sim \mathcal{D}^n}} \left[ \forall z_0 \in \mathcal{U}(x), \forall \tilde{z}_0 \in \mathcal{U}(\tilde{x}), \forall h \in \Delta_{\mathcal{H}}^{\mathcal{U}}(z_0, y, \tilde{z}_0) : \mathrm{err}_{\tilde{z}_0, \tilde{y}}(h) \leq \epsilon \right] \geq 1 - \delta \qquad (37)$$

as $\mathrm{OPT}_{\mathcal{U}^{-1}(\mathcal{U})} = 0$.

Suppose $(x, y), (\tilde{x}, \tilde{y}) \sim \mathcal{D}^n$. Now, let $z \in \mathcal{U}^3(x), \tilde{z} \in \mathcal{U}^3(\tilde{x})$ and take some $z_0 \in \mathrm{ip}_{\mathcal{U}, \mathcal{U}^2}(x, z), \tilde{z}_0 \in \mathrm{ip}_{\mathcal{U}, \mathcal{U}^2}(\tilde{x}, \tilde{z})$, both of which are necessarily nonempty as $\mathcal{U}^3 = \mathcal{U}^2 \mathcal{U}$, and $\hat{h} \in \mathrm{F}_{\mathcal{U}}\left(\Delta_{\mathcal{H}}^{\mathcal{U}}(z_0, y, \tilde{z}_0)\right)$.

Write $\hat{h} = \mathrm{F}_{\mathcal{U}}(h)$ for some $h \in \Delta_{\mathcal{H}}^{\mathcal{U}}(z_0, y, \tilde{z}_0)$.

From Equation (37) (replacing $z$ with $z_0$ and $\tilde{z}$ with $\tilde{z}_0$), it is enough to show that

$$\mathrm{err}^{\mathrm{rej}}(\hat{h}; x, y, \tilde{x}, \tilde{z}, \tilde{y}) \leq \mathrm{err}_{\tilde{z}_0, \tilde{y}}(h).$$

Suppose that $\hat{h}$ incurs an error under rejection at point $\tilde{z}_i$; it is enough to show that $h$ incurs an error at $\tilde{z}_{0_i}$. Furthermore, note that because $h \in \Delta_{\mathcal{H}}^{\mathcal{U}}(z_0, y, \tilde{z}_0)$, we have that $h(\mathcal{U}^{-1}(\tilde{z}_{0_i})) = \{h(\tilde{z}_{0_i})\}$ as $\tilde{z}_{0_i} \in \mathcal{U}^{-1}(\tilde{z}_{0_i})$. Write $h(\tilde{z}_{0_i}) = \hat{y}_i$.

We have one of the following:

1. $\hat{h}(\tilde{z}_i) \neq \tilde{y}_i$ and $\tilde{z}_i = \tilde{x}_i$

2. $\hat{h}(\tilde{z}_i) \notin \{\tilde{y}_i, \perp\}$ and $\tilde{z}_i \neq \tilde{x}_i$

In the first case, we must have $\tilde{z}_{0_i} = \tilde{x}_i$ as well as $\tilde{z}_{0_i}$ is an intermediate perturbation between $\tilde{x}_i$ and $\tilde{z}_i$, so, as $h(\mathcal{U}^{-1}(\tilde{z}_i)) = h(\mathcal{U}^{-1}(\tilde{z}_{0_i})) = \hat{y}_i$, $\hat{h}$ does not reject $\tilde{z}_{0_i}$ and $\hat{h}(\tilde{z}_{0_i}) = \hat{y}_i$. Hence, $h(\tilde{z}_{0_i}) = \hat{y}_i$ as well so, as $\hat{h}$ makes an error at $\tilde{z}_i$, $\hat{y}_i \neq y$ and so $h$ makes an error at $\tilde{z}_{0_i}$.

In the second case, if $h(\mathcal{U}^{-1}(\tilde{z}_i)) \neq \{h(\tilde{z}_i)\}$, then $\hat{h}$ would reject $\tilde{z}_i$ and hence not incur an error. So $h(\mathcal{U}^{-1}(\tilde{z}_i)) = \{h(\tilde{z}_i)\}$ and so $\hat{h}(\tilde{z}_i) = h(\tilde{z}_i)$. Since $\tilde{z}_{0_i} \in \mathcal{U}(\tilde{x}_i) \cap \mathcal{U}^{-2}(\tilde{z}_i)$, there exists some $\tilde{z}'_{0_i} \in \mathcal{U}(\tilde{z}_{0_i}) \cap \mathcal{U}^{-1}(\tilde{z}_i)$ and so, $h(\tilde{z}_{0_i}) = h(\tilde{z}'_{0_i}) = h(\tilde{z}_i) = \hat{h}(\tilde{z}_i) = \hat{y}_i$, so $h$ incurs an error at $\tilde{z}_{0_i}$.

In either case, we have that $h$ makes an error at $\tilde{z}_{0_i}$, showing the result. $\square$

**Sample Complexity**   Given $\epsilon$ and $\delta$, we need

$$\frac{\mathrm{rdim}_{\mathcal{U}^{-1}}(\mathcal{H})\log(2n) + \log(1/\delta)}{n} \leq \epsilon$$

for the result to hold.

Now, noting that $\log(2n) = 1 + \log n \leq 1 + \sqrt{n}$ for $n \geq 16$; hence we need to solve for the $n$ such that

$$\frac{\mathrm{rdim}_{\mathcal{U}^{-1}}(\mathcal{H})(1 + \sqrt{n}) + \log(1/\delta)}{n} = \epsilon$$

or, equivalently

$$\frac{\mathrm{rdim}_{\mathcal{U}^{-1}}(\mathcal{H}) + \log(\frac{1}{\delta}) + \sqrt{n}}{n} = \epsilon$$

or

$$\sqrt{n} = n\epsilon - \mathrm{rdim}_{\mathcal{U}^{-1}}(\mathcal{H}) - \log(\frac{1}{\delta})$$

or

$$n = n^2\epsilon^2 - 2\epsilon\left(\mathrm{rdim}_{\mathcal{U}^{-1}}(\mathcal{H}) + \log(\frac{1}{\delta})\right)n + \left(\mathrm{rdim}_{\mathcal{U}^{-1}}(\mathcal{H}) + \log(\frac{1}{\delta})\right)^2$$

or

$$n^2\epsilon^2 - \left(2\epsilon\left(\mathrm{rdim}_{\mathcal{U}^{-1}}(\mathcal{H}) + \log(\frac{1}{\delta})\right) + 1\right)n + \left(\mathrm{rdim}_{\mathcal{U}^{-1}}(\mathcal{H}) + \log(\frac{1}{\delta})\right)^2 = 0.$$

Solving, the result holds if

$$n \geq \frac{2\epsilon\left(\mathrm{rdim}_{\mathcal{U}^{-1}}(\mathcal{H}) + \log(\frac{1}{\delta})\right) + 1 + \sqrt{\left(2\epsilon\left(\mathrm{rdim}_{\mathcal{U}^{-1}}(\mathcal{H}) + \log(\frac{1}{\delta})\right) + 1\right)^2 - 4\left(\mathrm{rdim}_{\mathcal{U}^{-1}}(\mathcal{H}) + \log(\frac{1}{\delta})\right)^2\epsilon^2}}{2\epsilon^2}$$

$$= O\left(\frac{\mathrm{rdim}_{\mathcal{U}^{-1}}(\mathcal{H}) + \log(\frac{1}{\delta})}{\epsilon} + \frac{\sqrt{\mathrm{rdim}_{\mathcal{U}^{-1}}(\mathcal{H}) + \log(\frac{1}{\delta})}}{\epsilon^{\frac{3}{2}}}\right)$$

and, similarly, using

$$\frac{\mathrm{rdim}_{\mathcal{U}^{-1}}(\mathcal{H})\log(2n) + \log(1/\delta)}{n} \leq \frac{\mathrm{VC}(\mathcal{H})\log(2n) + \log(1/\delta)}{n}$$

we have the result if

$$n = O\left(\frac{\mathrm{VC}(\mathcal{H}) + \log(\frac{1}{\delta})}{\epsilon} + \frac{\sqrt{\mathrm{VC}(\mathcal{H}) + \log(\frac{1}{\delta})}}{\epsilon^{\frac{3}{2}}}\right)$$

**Remark:**   If $\mathrm{OPT}_{\mathcal{U}^{-1}\mathcal{U}} = 0$, we can guarantee the existence of an $\hat{h}$ which satisfies our conditions, but we can't guarantee that we will find it, as we cannot find $\Delta_{\mathcal{H}}^{\mathcal{U}}(z_0, y, \tilde{z}_0)$ without $z_0$ and $\tilde{z}_0$. We can, however, construct that an algorithm which, if it returns a model, always returns on which meets the conditions.

**Simplified Result**   To obtain a bound which does not involve an intermediate perturbation step, we may let

$$\Delta_{\mathrm{rej},\mathcal{H}}^{\mathcal{U}}(z, y, \tilde{z}) := \begin{cases} \hat{\Delta} \cup \Delta_{\mathrm{rej},\mathcal{H}}^{\mathcal{U}\prime}(z, y, \tilde{z}) & |\hat{\Delta}_{\mathcal{H}}^{\mathcal{U}}(z, y, \tilde{z})(\tilde{z})| = 1, \text{ and} \\ l\Delta_{\mathrm{rej},\mathcal{H}}^{\mathcal{U}\prime}(z, y, \tilde{z}) & \text{otherwise} \end{cases}$$

where

$$\Delta_{\mathrm{rej},\mathcal{H}}^{\mathcal{U}\prime}(z, y, \tilde{z}) = \bigcap_{\tilde{z}' \in \mathcal{U}^{-2}(\tilde{z})} \Delta_{\mathcal{H}}^{\mathcal{U}}(z, y, \tilde{z}')$$

where

$$\hat{\Delta}_{\mathcal{H}}^{\mathcal{U}}(z, y, \tilde{z}) = \bigcup_{\tilde{z}' \in \mathcal{U}^{-2}(\tilde{z})} \Delta_{\mathcal{H}}^{\mathcal{U}}(z, y, \tilde{z}').$$

If $|\hat{\Delta}^{\mathcal{U}}_{\mathcal{H}}(z, y, \tilde{z}_0)(\tilde{z})| = 1$, then as $\Delta^{\mathcal{U}}_{\mathcal{H}}(z_0, y, \tilde{z}_0)(\tilde{z}) \subseteq \hat{\Delta}^{\mathcal{U}}_{\mathcal{H}}(z, y, \tilde{z})(\tilde{z})$, $\hat{\Delta}^{\mathcal{U}}_{\mathcal{H}}(z, y, \tilde{z})(\tilde{z}) = \Delta^{\mathcal{U}}_{\mathcal{H}}(z_0, y, \tilde{z}_0)(\tilde{z})$ since $\Delta^{\mathcal{U}}_{\mathcal{H}}(z_0, y, \tilde{z}_0)$ is nonempty as $\mathrm{OPT}_{\mathcal{U}^{-1}(\mathcal{U})} = 0$.

Note that for common classes of perturbations, we can simplify the $\Delta'_{\mathrm{rej}}$. Note that the conditions of the theorem hold for perturbations defined via $\epsilon$-balls in a metric.

**Lemma A.10.** *In the realizable case, if $\mathcal{U} = \mathcal{U}^{-1}$,*

$$\Delta^{\mathcal{U}'}_{\mathrm{rej},\mathcal{H}}(z, y, \tilde{z}) = \Delta^{\mathcal{U}^3}_{\mathcal{H}}(z, y, \tilde{z})$$

*Proof.* Suppose $h \in \Delta^{\mathcal{U}'}_{\mathrm{rej},\mathcal{H}}(z, y, \tilde{z})$. Then by the definitions of $\Delta_{\mathrm{rej}}$ and $\Delta$, for any $z' \in \mathcal{U}^{-2}(z)$, $\tilde{z}' \in \mathcal{U}^{-2}(\tilde{z})$, we have that, for any $x \in \mathcal{U}^{-1}(z')$ and $\tilde{x} \in \mathcal{U}^{-1}(\tilde{z}')$, $h(x_i) = h(z'_i)$ and $h(\tilde{x}_i) = h(\tilde{z}'_i)$. Now, as there exists some $z'' \in \mathcal{U}(z') \cap \mathcal{U}^{-1}(bz)$ and $h(x) = h(z') = h(z'') = h(z)$ by an argument similar to that in Theorem A.9 and similarly for $\tilde{x}$ and $\tilde{z}$, we have that for any $x \in \mathcal{U}^{-3}(z)$ and $\tilde{x} \in \mathcal{U}^{-3}(\tilde{z})$, $h(x_i) = h(z_i)$ and $h(\tilde{x}_i) = h(\tilde{z}_i)$, and so

$$\Delta^{\mathcal{U}'}_{\mathrm{rej},\mathcal{H}}(z, y, \tilde{z}) \subseteq \Delta^{\mathcal{U}^3}_{\mathcal{H}}(z, y, \tilde{z})$$

Now, if $h \in \Delta^{\mathcal{U}^3}_{\mathcal{H}}(z, y, \tilde{z})$, we have that, for any $x \in \mathcal{U}^{-3}(z)$ and $\tilde{x} \in \mathcal{U}^{-3}(\tilde{z})$, $h(x_i) = h(z_i)$ and $h(\tilde{x}_i) = h(\tilde{z}_i)$. Now, suppose $z' \in \mathcal{U}^{-2}(z)$, $\tilde{z}' \in \mathcal{U}^{-2}(\tilde{z})$. Since $x \in \mathcal{U}(x)$ for all $x$, $z' \in \mathcal{U}^{-3}(z)$, $\tilde{z}' \in \mathcal{U}^{-3}(\tilde{z})$ as well. Hence, $h(z'_i) = h(z_i)$ and $h(\tilde{z}'_i) = h(\tilde{z}_i)$. Now, if $x \in \mathcal{U}^{-1}(z')$ and $\tilde{x} \in \mathcal{U}^{-1}(\tilde{z}')$, we have $x \in \mathcal{U}^{-3}(z)$ and $\tilde{x} \in \mathcal{U}^{-3}(\tilde{z})$ and so $h(x_i) = h(z_i)$ and $h(\tilde{x}_i) = h(\tilde{z}_i)$. But then $h(x_i) = h(z'_i)$ and $h(\tilde{x}_i) = h(\tilde{z}'_i)$. Hence, we have that

$$\Delta^{\mathcal{U}^3}_{\mathcal{H}}(z, y, \tilde{z}) \subseteq \Delta^{\mathcal{U}'}_{\mathrm{rej},\mathcal{H}}(z, y, \tilde{z})$$

and the result follows. $\qquad\square$

From this, we immediately derive the corollary

$$\Delta^{\mathcal{U}}_{\mathrm{rej},\mathcal{H}}(z, y, \tilde{z}) \supseteq \Delta^{\mathcal{U}^3}_{\mathcal{H}}(z, y, \tilde{z}).$$

**Remark:** Note that this means that $\Delta^{\mathcal{U}'}_{\mathrm{rej},\mathcal{H}}(z, y, \tilde{z})$ is nonempty if $\mathrm{OPT}_{\mathcal{U}^6} = 0$, and, by the definition of $\Delta$, $\Delta$ is also nonempty if $|\hat{\Delta}^{\mathcal{U}}_{\mathcal{H}}(z, y, \tilde{z})(\tilde{z})| = 1$, i.e. if there exists only one possible labeling of the $\tilde{z}$ which is robust at some possible intermediate perturbation.

Now, by the above and from Theorem A.9 we may immediately derive Theorem 4.1 by noting that if $\mathcal{U} = \mathcal{U}^{-1}$, $\mathcal{U}^{-1}\mathcal{U} = \mathcal{U}^2$, and if $\hat{h} \in F_{\mathcal{U}}(\Delta^{\mathcal{U}}_{\mathcal{H}}(z, y, \tilde{z})) = F_{\mathcal{U}^{1/3}}(\Delta^{\mathcal{U}^{1/3}}_{\mathrm{rej},\mathcal{H}}(z, y, \tilde{z}))$ then we have $\hat{h} \in F_{\mathcal{U}^{1/3}}\left(\Delta^{\mathcal{U}^{1/3}}_{\mathcal{H}}(z_0, y, \tilde{z}_0)\right)$ for some $z_0 \in \mathrm{ip}_{\mathcal{U}^{1/3}, \mathcal{U}^{2/3}}(x, z)$ and $\tilde{z}_0 \in \mathrm{ip}_{\mathcal{U}^{1/3}, \mathcal{U}^{2/3}}(\tilde{x}, \tilde{z})$.

Furthermore, following from Lemma A.10, $\Delta^{\mathcal{U}^{1/3}}_{\mathrm{rej},\mathcal{H}}(z, y, \tilde{z})$ is nonempty is $OPT_{\mathcal{U}^2} = 0$, showing completeness that the $\Delta$ of Theorem 4.1 is nonempty under that condition, as well as, as noted above, under the condition that there exists only one possible labeling consistent on a potential intermediate perturbation.

Now, we demonstrate that there exists a data distribution for which the transductive learner implied by $\Delta$ finds a solution for which the bound applies, but where no transductive learner has zero asymptotic robust error

**Theorem A.11.** *There exists a distribution $\mathcal{D}$ over $\mathcal{X} \times \mathcal{Y}$, a hypothesis class $\mathcal{H}$, and perturbation set $\mathcal{U}$ for which, with probability $\geq 1 - 2^{1-n}$, for any $(x, y), (\tilde{x}, \tilde{y}) \sim \mathcal{D}^n$ and any $\tilde{z} \in \mathcal{U}^3(\tilde{z})$, $\Delta^{\mathcal{U}}_{\mathrm{rej},\mathcal{H}}(x, y, \tilde{z})$ is nonempty and for all $h \in \Delta^{\mathcal{U}}_{\mathrm{rej},\mathcal{H}}(z, y, \tilde{z})$, $\mathrm{err}^{\mathrm{rej}}_{\mathcal{U}}(h; x, y, \tilde{x}, \tilde{z}, \tilde{y}) = 0$ but, there exists no transductive learner (without rejection) $\mathbb{A}$ for which $\lim_{n\to\infty} \mathbb{E}\left[\sup_{\tilde{z} \in \mathcal{U}(\tilde{x})} \mathrm{err}_{\mathcal{U}}(\mathbb{A}(x, y, \tilde{z}); x, y, \tilde{z}, \tilde{y})\right] < 1/2.$*

*Proof.* Consider the simple discrete distribution $\mathcal{D}$ with $(x, y) \sim \mathcal{D}$ is $(1, 1)$ with probability $1/2$ and $(-1, 0)$ with probability $1/2$. Now, let $\mathcal{U}(x) = \{y \mid |y - x| < 1.5\}$ and let $\mathcal{H}$ be the class of

Now, let $\mathcal{H}$ be the class of threshold functions $h_w(x) = \mathbb{1}_{x \geq w}$ and $h_w^-(x) = \mathbb{1}_{x < w}$ for integer $w$.

First, note that with probability $1 - 2^{1-n}$ both $(-1, 0)$ and $(1, 1)$ appear in $\boldsymbol{x}$. In that case, any $h \in \Delta_{\mathcal{H}}^{\mathcal{U}}(\boldsymbol{x}, \boldsymbol{y}, \tilde{z}')$ must be robust at $-1$ and $1$ up to a radius of $1/2$; and hence $h$ must be $h_w$ for some $w \in [-1/2, 1/2]$ (and hence, $w = 0$). Hence, $|\hat{\Delta}| \leq 1$; note that for any possible perturbation of $-1$ or $1$ is within $\mathcal{U}^2$ (i.e. within 1 unit of) either $-1$ or $1$; hence, there always exists some $\tilde{z}'$ where $\Delta_{\mathcal{H}}^{\mathcal{U}}(\boldsymbol{x}, \boldsymbol{y}, \tilde{z}')$ is nonempty.

But then, there must exist exactly one element in $\hat{\Delta}$, and so $\Delta$ is nonempty. Consider $\tilde{z}_i$. We have two cases:

If $\tilde{z}_i \in [-1, -1/2] \cup [1/2, 1]$, then, as $h$ is robustly correct with radius $1/2$ about $1$ and $-1$, then $\tilde{x}_i = \text{sign}(\tilde{z}_i)$ and hence $h(\tilde{x}_i) = \text{sign}(\tilde{z}_i)$. If $\tilde{x}_i = \tilde{z}_i$ we do not reject as $h$ is robust with radius $1/2$ about $-1$ and $1$. Thus, we do not incur an error at $\tilde{z}_i$.

If $\tilde{z}_i \in (-1/2, 1/2)$, then $\tilde{z}_i$ must be perturbed. But we have both positive and negative values within $1/2$ of $\tilde{z}_i$, and so $F_{\mathcal{U}}(\tilde{z}_i) = \perp$. Hence, we do not occur an error at $\tilde{z}_i$.

In all cases, we do not incur an error if both $x = -1$ and $x = 1$ appear in the training data, and so $\text{err}_{\mathcal{U}}^{\text{rej}}(h; \boldsymbol{x}, \boldsymbol{y}, \tilde{x}, \tilde{z}, \tilde{y})$ is $0$ with probability $\geq 1 - 2^{1-n}$.

To see that there exists no transductive algorithm (without rejection) that can have asympotic error below $1/2$, note that any $\tilde{x}$ can be perturbed to $\tilde{z}$ where all $\tilde{z}$ are $0$; hence, samples from class $0$ and class $1$ are indistinguishable and the minimum error on $\tilde{z}$ achievable by $h$ is the minimum of the fraction of the $\tilde{x}$ which are $-1$ and the fraction which are $1$. As $n \to \infty$, these both tend to $1/2$ and the result follows. $\qquad\square$

## A.4 TRANSDUCTION+REJECTION: AGNOSTIC CASE

Note that, if $\mathcal{U}$ can be decomposed into a form $\mathcal{U} = (\mathcal{U}^{1/3})^3$ where $\mathcal{U}^{1/3} = \mathcal{U}^{-1/3}$ (as with standard perturbations in $l_p$), we obtain a bound which depends on $\text{OPT}_{\mathcal{U}^{2/3}}$ rather than $\text{OPT}_{\mathcal{U}^2}$, enabling, for $\hat{h}$ satisfying the conditions, much stronger guarantees if $\text{OPT}_{\mathcal{U}^{2/3}} << \text{OPT}_{\mathcal{U}^2}$. Note that as $\forall x \, x \in \mathcal{U}(x)$, $\forall x \, \mathcal{U}^{2/3}(x) \subseteq \mathcal{U}^2(x)$, and so $\text{OPT}_{\mathcal{U}^{2/3}} \leq \text{OPT}_{\mathcal{U}^2}$.

**Theorem A.12.** *For any $n \in \mathbb{N}$, $\delta > 0$, class $\mathcal{H}$, perturbation set $\mathcal{U}$, and distribution $\mathcal{D}$ over $\mathcal{X} \times \mathcal{Y}$:*

$$\Pr_{\substack{(\boldsymbol{x},\boldsymbol{y}) \sim \mathcal{D}^n \\ (\tilde{x},\tilde{y}) \sim \mathcal{D}^n}} \left[ \begin{array}{l} \forall z \in \mathcal{U}^3(\boldsymbol{x}), \forall z_0 \in \text{ip}_{\mathcal{U},\mathcal{U}^2}(\boldsymbol{x}, z), \forall \tilde{z} \in \mathcal{U}^3(\tilde{x}), \forall \tilde{z}_0 \in \text{ip}_{\mathcal{U},\mathcal{U}^2}(\tilde{x}, \tilde{z}), \\ \forall \hat{h} \in F_{\mathcal{U}}\left(\Delta_{\mathcal{H}}^{\mathcal{U}}(z_0, \boldsymbol{y}, \tilde{z}_0)\right) : \text{err}^{\text{rej}}(\hat{h}; \boldsymbol{x}, \boldsymbol{y}, \tilde{x}, \tilde{z}, \tilde{y}) \leq \epsilon \end{array} \right] \geq 1 - \delta$$

*where*

$$\epsilon = \min \left\{ 2\,\text{OPT}_{\mathcal{U}^{-1}\mathcal{U}} + O\left(\sqrt{\frac{\text{VC}(\mathcal{H}) + \log(1/\delta)}{n}}\right), 3\text{OPT}_{\mathcal{U}^{-1}\mathcal{U}} + O\left(\sqrt{\frac{\text{rdim}\,\mathcal{U}(\mathcal{H}) \ln(2n) + \ln(1/\delta)}{n}}\right) \right\}.$$

*Proof.* Suppose $(\boldsymbol{x}, \boldsymbol{y}), (\tilde{x}, \tilde{y}) \sim \mathcal{D}^n$. Now, let $z \in \mathcal{U}^3(\boldsymbol{x}), \tilde{z} \in \mathcal{U}^3(\tilde{x})$ and take some $z_0 \in \text{ip}_{\mathcal{U},\mathcal{U}^2}(\boldsymbol{x}, z), \tilde{z}_0 \in \text{ip}_{\mathcal{U},\mathcal{U}^2}(\tilde{x}, \tilde{z})$, both of which are necessarily nonempty, and $\hat{h} \in F_{\mathcal{U}}\left(\Delta_{\mathcal{H}}^{\mathcal{U}}(z_0, \boldsymbol{y}, \tilde{z}_0)\right)$.

Write $\hat{h} = F_{\mathcal{U}}(h)$ for some $h \in \Delta_{\mathcal{H}}^{\mathcal{U}}(z_0, \boldsymbol{y}, \tilde{z}_0)$.

We will begin as in Theorem A.9. As before, there are two cases in which $\hat{h}$ can incur an error at $\tilde{z}_i$:

1. $\hat{h}(\tilde{z}_i) \neq \tilde{y}_i$ and $\tilde{z}_i = \tilde{x}_i$
2. $\hat{h}(\tilde{z}_i) \notin \{\tilde{y}_i, \perp\}$ and $\tilde{z}_i \neq \tilde{x}_i$

Now, if $\tilde{z}_i = \tilde{x}_i$, an error occurs if $\hat{h}$ rejects $\tilde{z}_i$ or if $h$ robustly predicts some $\hat{y}_i \neq \tilde{y}_i$; hence an error occurs if $h$ is not $\mathcal{U}^{-1}$-robust at $\tilde{z}_{0_i}$ or if $h(\tilde{z}_{0_i}) \neq \tilde{y}_i$.

Otherwise, $h$ must be $\mathcal{U}^{-1}$-robust at $\tilde{z}_i$, as, otherwise, $\hat{h}$ would reject $\tilde{z}_i$. Hence, as there exists some $\tilde{z}_{0_i}' \in \mathcal{U}(\tilde{z}_{0_i}) \cap \mathcal{U}^{-1}(\tilde{z}_i)$, if $h$ is $\mathcal{U}$-robust at $\tilde{z}_{0_i}$, we must have $h(\tilde{z}_i) = h(\tilde{z}_{0_i})$, and so, if $\hat{h}$ makes an error, $h$ is not $\mathcal{U}^{-1}$-robust at $\tilde{z}_{0_i}$ or $h(\tilde{z}_{0_i}) \neq \tilde{y}_i$.

Now, in both cases, errors only occur if $h$ is not $\mathcal{U}^{-1}$-robust at $\tilde{z}_{0_i}$ or $h(\tilde{z}_{0_i}) \neq \tilde{y}_i$. As $\tilde{x}_i \in \mathcal{U}^{-1}(\tilde{z}_{0_i})$, we have, equivalently, that an error occurs if $h$ is not $\mathcal{U}^{-1}$-robust at $\tilde{z}_{0_i}$ or $h(\tilde{x}_i) \neq \tilde{y}_i$.

Hence,

$$\text{err}^{\text{rej}}(\hat{h}; \boldsymbol{x}, \boldsymbol{y}, \tilde{\boldsymbol{x}}, \tilde{\boldsymbol{z}}, \tilde{\boldsymbol{y}}) \leq \text{err}^{\text{rej}}(h; \tilde{x}, \tilde{y}) + \text{R}_{\mathcal{U}^{-1}}(h; \tilde{\boldsymbol{z}}_0)$$

Now, the right hand is exactly what is bounded in Theorem 2 of Montasser et al. (2021); as we have $h \in \Delta_{\mathcal{H}}^{\mathcal{U}}(\boldsymbol{z}_0, \boldsymbol{y}, \tilde{\boldsymbol{z}}_0)$, we have

$$\text{err}^{\text{rej}}(\hat{h}; \boldsymbol{x}, \boldsymbol{y}, \tilde{\boldsymbol{x}}, \tilde{\boldsymbol{z}}, \tilde{\boldsymbol{y}}) \leq \text{err}^{\text{rej}}(h; \tilde{x}, \tilde{y}) + \text{R}_{\mathcal{U}^{-1}}(h; \tilde{\boldsymbol{z}}_0) \leq \epsilon$$

where

$$\epsilon = \min\left\{ 2\,\text{OPT}_{\mathcal{U}^{-1}\mathcal{U}} + O\left( \sqrt{\frac{\text{VC}(\mathcal{H}) + \log(1/\delta)}{n}} \right), 3\text{OPT}_{\mathcal{U}^{-1}\mathcal{U}} + O\left( \sqrt{\frac{\text{rdim}\,\mathcal{U}(\mathcal{H})\ln(2n) + \ln(1/\delta)}{n}} \right) \right\}$$

with probability $\geq 1 - \delta$ by its proof. $\qquad\square$

As in the realizable case, we can immediately derive the following corollary. However, we cannot simplify the definition of $\Delta_{\text{rej}}$ as before; see Lemma A.14.

**Corollary A.13.** *For any $n \in \mathbb{N}$, $\delta > 0$, class $\mathcal{H}$, perturbation set $\mathcal{U}$ where $\mathcal{U} = \mathcal{U}^{-1}$, and distribution $\mathcal{D}$ over $\mathcal{X} \times \mathcal{Y}$:*

$$\Pr_{\substack{(\boldsymbol{x},\boldsymbol{y})\sim\mathcal{D}^n \\ (\tilde{\boldsymbol{x}},\tilde{\boldsymbol{y}})\sim\mathcal{D}^n}} \left[ \begin{array}{l} \forall \boldsymbol{z} \in \mathcal{U}^3(\boldsymbol{x}), \forall \tilde{\boldsymbol{z}} \in \mathcal{U}^3(\tilde{\boldsymbol{x}}), \forall \hat{h} \in \text{F}_{\mathcal{U}}\left( \Delta_{\text{rej},\mathcal{H}}^{\mathcal{U}}(\boldsymbol{z}, \boldsymbol{y}, \tilde{\boldsymbol{z}}) \right): \\ \text{err}^{\text{rej}}(\hat{h}; \boldsymbol{x}, \boldsymbol{y}, \tilde{\boldsymbol{x}}, \tilde{\boldsymbol{z}}, \tilde{\boldsymbol{y}}) \leq \epsilon \end{array} \right] \geq 1 - \delta$$

*where*

$$\epsilon = \min\left\{ 2\,\text{OPT}_{\mathcal{U}^{-1}\mathcal{U}} + O\left( \sqrt{\frac{\text{VC}(\mathcal{H}) + \log(1/\delta)}{n}} \right), 3\text{OPT}_{\mathcal{U}^{-1}\mathcal{U}} + O\left( \sqrt{\frac{\text{rdim}\,\mathcal{U}(\mathcal{H})\ln(2n) + \ln(1/\delta)}{n}} \right) \right\}.$$

**Lemma A.14.** *In the agnostic case, we have that if $\mathcal{U} = \mathcal{U}^{-1}$,*

$$\Delta_{\text{rej},\mathcal{H}}^{\mathcal{U}}(\boldsymbol{z}, \boldsymbol{y}, \tilde{\boldsymbol{z}}) \subseteq \Delta_{\mathcal{H}}^{\mathcal{U}^3}(\boldsymbol{z}, \boldsymbol{y}, \tilde{\boldsymbol{z}})$$

*Proof.* By the definition of R, we have

$$\text{R}_{\mathcal{U}^{-3}}(h; \tilde{\boldsymbol{z}}) = \frac{1}{n} \sum_{i=1}^{n} \mathbb{1}\left\{ \exists \tilde{x}_i \in \mathcal{U}^{-3}(\tilde{z}_i) : h(\tilde{x}_i) \neq h(\tilde{z}_i) \right\}$$

$$= \frac{1}{n} \sum_{i=1}^{n} \mathbb{1}\left\{ \exists \tilde{z}_i' \in \mathcal{U}^{-2}(\tilde{z}_i)\, \exists \tilde{x}_i \in \mathcal{U}^{-1}(\tilde{z}_i') : h(\tilde{x}_i) \neq h(\tilde{z}_i) \right\}$$

$$= \max_{\tilde{z}_i' \in \mathcal{U}^{-2}(\tilde{z}_i)} \frac{1}{n} \sum_{i=1}^{n} \mathbb{1}\left\{ \exists \tilde{x}_i \in \mathcal{U}^{-1}(\tilde{z}_i') : h(\tilde{x}_i) \neq h(\tilde{z}_i) \right\}$$

$$= \max_{\tilde{z}_i' \in \mathcal{U}^{-2}(\tilde{z}_i)} \text{R}_{\mathcal{U}^{-1}}(h; \tilde{\boldsymbol{z}}')$$

where the last equality holds as $x \in \mathcal{U}(x)$ for all $x$ and as $\mathcal{U} = \mathcal{U}^{-1}$, which together show that if for some $\tilde{z}_i$ and $\tilde{z}_i' \in \mathcal{U}^{-2}(\tilde{z}_i)$ we have that $h(\tilde{z}_i') \neq h(\tilde{z}_i)$, that either there exists some $\tilde{z}_i'' \in \mathcal{U} = \mathcal{U}^{-1}(\tilde{z}_i')$ such that $h(\tilde{z}_i'') \neq h(\tilde{z}_i')$ or there exists some $\tilde{z}_i'' \in \mathcal{U} = \mathcal{U}^{-1}(\tilde{z}_i)$ such that $h(\tilde{z}_i'') \neq h(\tilde{z}_i)$ (as before, note that $\tilde{z}_i = \mathcal{U}(\tilde{z}_i'')$ for some $\tilde{z}_i''\in\mathcal{U}(\tilde{z}_i')$ by the definition of $\mathcal{U}^3$); the reverse is similar.

We can derive a result for $\text{R}_{\mathcal{U}^{-3}}(h; \boldsymbol{z}, \boldsymbol{y})$ similarly.

Suppose $h \in \Delta_{\text{rej},\mathcal{H}}^{\mathcal{U}}(\boldsymbol{z}, \boldsymbol{y}, \tilde{\boldsymbol{z}})$. Then, $h$ minimizes $\max\{\text{R}_{\mathcal{U}^{-1}}(h; \boldsymbol{z}', \boldsymbol{y}), \text{R}_{\mathcal{U}^{-1}}(h; \tilde{\boldsymbol{z}}')\}$ for all $\boldsymbol{z}' \in \mathcal{U}^{-2}(\boldsymbol{z})$, $\tilde{\boldsymbol{z}}' \in \mathcal{U}^{-2}(\tilde{\boldsymbol{z}})$, so by the above, $h$ must also minimize

$$\max_{\boldsymbol{z}' \in \mathcal{U}^{-2}(\boldsymbol{z}), \tilde{\boldsymbol{z}}' \in \mathcal{U}^{-2}(\tilde{\boldsymbol{z}})} \max\{\text{R}_{\mathcal{U}^{-1}}(h; \boldsymbol{z}', \boldsymbol{y}), \text{R}_{\mathcal{U}^{-1}}(h; \tilde{\boldsymbol{z}}')\}$$

$$= \max\left\{ \max_{\boldsymbol{z}' \in \mathcal{U}^{-2}(\boldsymbol{z})} \text{R}_{\mathcal{U}^{-1}}(h; \boldsymbol{z}', \boldsymbol{y}), \max_{\tilde{\boldsymbol{z}}' \in \mathcal{U}^{-2}(\tilde{\boldsymbol{z}})} \text{R}_{\mathcal{U}^{-1}}(h; \tilde{\boldsymbol{z}}') \right\}$$

$$= \max\{\text{R}_{\mathcal{U}^{-3}}(h; \tilde{\boldsymbol{z}}), \text{R}_{\mathcal{U}^{-3}}(h; \boldsymbol{z}, \boldsymbol{y})\}$$

and so $h \in \Delta_{\mathcal{H}}^{\mathcal{U}^3}(z, y, \tilde{z})$.

However, minimizing

$$\max_{z' \in \mathcal{U}^{-2}(z), \tilde{z}' \in \mathcal{U}^{-2}(\tilde{z})} \max \left\{ R_{\mathcal{U}^{-1}}(h; z', y), R_{\mathcal{U}^{-1}}(h; \tilde{z}') \right\}$$

does not necessarily imply that $h$ minimizes $\max \left\{ R_{\mathcal{U}^{-1}}(h; z', y), R_{\mathcal{U}^{-1}}(h; \tilde{z}') \right\}$ for all $z' \in \mathcal{U}^{-2}(z), \tilde{z}' \in \mathcal{U}^{-2}(\tilde{z})$, so the reverse may not hold. $\qquad\square$

## A.5 Extension to Unbalanced Training and Test Data

We provide a sketch of a proof that allows extending Theorem 1 of (Montasser et al., 2021) to unbalanced training and test sets; however, for simplicity, we will work with the original form. The assumptions are the same, except that we have $n$ training points and $m$ test points.

The proof is exactly as before up to the "Finite robust labelings" portion (which points are and are not labelled don't matter up to then and the symmetry arguments still apply). The basic idea of determining the probability of zero loss on the training and test sets and error $> \epsilon$ on the test examples with permutation still applies. Let $E_{\sigma, x}$ be the event that there exists a labelling $\hat{h}(x_{\sigma(1:n+m)})$ in the allowable set where this occurs.

We have

$$\Pr_{\sigma}\left[E_{\sigma, x}\right] \le \Pr_{\sigma}\left[\exists \hat{h} \in \Pi_{\mathcal{H}}^{\mathcal{U}}(x_1, \ldots, x_{n+m}) : \mathrm{err}_{x_{\sigma(1:n)}, y_{\sigma(1:n)}}(\hat{h}) = 0 \wedge \mathrm{err}_{x_{\sigma(n:n+m)}, y_{\sigma(n:n+m)}}(\hat{h}) > \epsilon\right]$$

and, as in (Montasser et al., 2021), note the probability of choosing such a perturbation $\sigma$ for a fixed $\hat{h}$ is at most

$$\left(\frac{m}{n+m}\right)^s \le \left(\frac{m}{n+m}\right)^{\lceil \epsilon m \rceil} = \left(\frac{n+m}{m}\right)^{-\lceil \epsilon m \rceil} \le \left(\frac{n+m}{m}\right)^{-\epsilon m}$$

if we assume the number of total errors $s \ge \lceil \epsilon m \rceil$ without loss of generality (otherwise, err $> \epsilon$ would be impossible).

Hence, by a union bound,

$$\Pr_{\sigma}\left[E_{\sigma, x}\right] \le \left|\Pi_{\mathcal{H}}^{\mathcal{U}}(x_1, \ldots, x_{n+m})\right| \left(\frac{n+m}{m}\right)^{-\epsilon m}$$

and so

$$\Pr_{\sigma}\left[E_{\sigma, x}\right] \le (n+m)^{\mathrm{rdim}_{\mathcal{U}^{-1}}(\mathcal{H})} \left(\frac{n+m}{m}\right)^{-\epsilon m}$$

by Sauer's Lemma (in the form of Lemma 3 of (Montasser et al., 2021)).

Now, we bound the probability by $\delta$, we need

$$(n+m)^{\mathrm{rdim}_{\mathcal{U}^{-1}}(\mathcal{H})} \left(\frac{n+m}{m}\right)^{-\epsilon m} \le \delta$$

which, solving, gives us

$$\epsilon \ge \frac{\mathrm{rdim}_{\mathcal{U}^{-1}}(\mathcal{H}) \log_{\frac{n+m}{m}}(n+m) + \log_{\frac{n+m}{m}} \frac{1}{\delta}}{m} = \frac{\mathrm{rdim}_{\mathcal{U}^{-1}}(\mathcal{H}) \log(n+m) + \log \frac{1}{\delta}}{m \log\left(1 + \frac{m}{n}\right)}$$

Which reduces to the original result if $n = m$ (note that the logarithms are base-2).

**Corollary** If we fix $n + m$, $\mathcal{H}$, and $\delta$, the guarantee is strongest (i.e. we minimize $\epsilon$) when $n = m$. To see this, consider the denominator. Write $\alpha = \frac{m}{n}$. Then, we wish to maximize $n\alpha \log(1 + \alpha)$ (or equivalently $f(\alpha) = \alpha \log(1 + \alpha)$ subject to $\alpha \ge 0$. Now, note that $f'(\alpha) = \log(1 + \alpha) - 1 = 0$ when $\alpha = 1$, i.e. when $m = n$.

Also, we can see from the result above, that if we fix $m$ and $\delta$, then the minimum value of $\epsilon$ tends towards $\infty$ as $n \to \infty$, so there does not necessarily exist a labelled training set sampled from $\mathcal{D}$ which provides a guarantee with high probability of arbitrarily low error on a fixed test set.

## B EXPERIMENTAL DETAILS

### B.1 COMPUTING INFRASTRUCTURE

We used a SLURM cluster with A100 GPUs to run our experiments.

### B.2 BASELINE DETAILS

The baselines are trained with standard adversarial training (Goodfellow et al., 2015) (Madry et al., 2018). Attacks against AT without rejection use standard PGD with a cross-entropy objective, while attacks against AT with rejection use PGD targeting $\mathcal{L}_{\text{REJ}}$ as described in algorithm 3. In all cases, the parameters for PGD in training are the same as those used in TLDR's training process for the same dataset.

### B.3 DEFENSE

In our implementation, we begin to incorporate the transductive term in our objective (see Equation (1)) after initially training the model with the inductive loss term only; this allows learning a better baseline before we begin to enforce robustness about the test points. In our experiments, we use the transductive loss in the final half of the training epochs.

### B.4 ADAPTIVE ATTACK

Solving for the perturbation $\tilde{x}$ by iteratively optimizing $\mathcal{L}_{\text{REJ}}$ poses several difficulties.

First, the rejection-avoidance term $\left\| \tilde{x} - \arg\max_{\|x' - \tilde{x}\| \le \epsilon} \mathcal{L}_{\text{DB},h}(x') \right\|$ is not differentiable with respect to $\tilde{x}$. While it is possible to approximate the derivative with the derivative of a proxy (e.g. differentiating though some fixed number of PGD steps, necessitating second-order optimization), this is extremely expensive and does not improve results in our experiments (see below).

Intuitively, we might see that this would be the case: if the decision boundary is smooth, we might expect the maximizers in $\mathcal{U}(x + \Delta)$ and $\mathcal{U}(x)$ to be the same for small $\Delta$ unless $x'$ is near the border of $\mathcal{U}(x)$ given that $\mathcal{U}(x + \Delta) \approx \mathcal{U}(x)$. In this case, approximating $x'$ as constant with respect to $x$ is reasonable.

In addition, note that if $h(x) = y$, the adversary must find a $\tilde{x}$ where $h(\tilde{x}) \neq y$ which is not rejected: if maximizing $\mathcal{L}_{\text{REJ}}$ with PGD, the rejection-avoidance term penalizes moving $\tilde{x}$ towards the decision boundary. As this is necessary to find a valid attack (when $h(\tilde{x}) = y$ at initialization), we adjust $\lambda$ adaptively during optimization by setting it to zero when $h(\tilde{x}) = y$.

### B.5 TRANSDUCTIVE ATTACK DETAILS

We present two rejection-aware transductive attacks: a stronger but more computationally intensive rejection-aware GMSA (Algorithm 1) and a weaker but faster rejection-aware transfer attack which takes the transductive robust rejection risk into account (Algorithm 2).

Finally, note the attack with $\mathcal{L}_{\text{REJ}}$, without GMSA, is effective against selective classifiers based on the transformation $F$ (and via Tramèr's equivalency, selective classifiers in general). So we summarize this attack on a fixed model in Algorithm 3.

### B.6 REJECTRON EXPERIMENTS

Goldwasser et al.'s implementation of Rejectron (Goldwasser et al., 2020) trains a classifier (call it $h_c$) on the training set and a discriminator ($h_d$) to distinguish between the (clean) training and (potentially-perturbed) test data. Samples are rejected if the discriminator classifies them as test data; otherwise, the classifier's prediction is returned. Our adaptive attack is then very simple: we follow the approach of Algorithm 1 but with a loss function $\mathcal{L}_{\text{DISC}}$ which targets the defense.

---

**Algorithm 1** REJECTION-AWARE GMSA

---

**Require:** A clean training set $T$, a clean test set $E$, a transductive learning algorithm for classifiers $\mathbb{A}$, an adversarial budget of $\epsilon$, *mode* either MIN or AVG, a radius used for rejection $\epsilon_{\text{defense}}$, and a maximum number of iterations $N \geq 1$. $E|_X$ refers to the projection on the feature space for $E$.

1: Search for a perturbation of the test set which fools the model space induced by $(T, \mathcal{U}(E|_X))$.
2: $E' = E$
3: $\hat{E} = E$
4: $\text{err}_{\max} = -\inf$
5: **for** i=0,...,N-1 **do**
6:     Train a transductive model on the perturbed data.
7:     $h^{(i)} = \mathbb{A}(T, E'|_X)$
8:

$$\text{err} = \frac{1}{|E'|} \sum_{i=1}^{|E'|} \mathbb{1} \left\{ \left( F(h^{(i)})(\tilde{x}_i) \notin \{\tilde{y}_i\} \wedge \tilde{x}_i = x_i \right) \vee \left( F(h^{(i)})(\tilde{x}_i) \notin \{\tilde{y}_i, \perp\} \wedge \tilde{x}_i \neq x_i \right) \right\}$$

    {The $\tilde{x}_i$ and the $x_i$ are the i$^{\text{th}}$ datapoints of $E'$ and $E$, repectively; $y_i$ is the true label.}
9:     **if** $\text{err}_{\max} < \text{err}$ **then**
10:         $\hat{E} = E'$
11:     **end if**
12:     **for** $j = 1, \ldots, |E|$ **do**
13:         **if** *mode* = MIN **then**
14:

$$\tilde{x}_j = \arg \max_{\|\tilde{x} - x_j\| \leq \epsilon} \min_{1 \leq k \leq i} \mathcal{L}_{\text{REJ}h^{(k)}}(\tilde{x}, y_j)$$

15:         **else**
16:

$$\tilde{x}_j = \arg \max_{\|\tilde{x} - x_j\| \leq \epsilon} \frac{1}{i} \sum_{k=1}^{i} \mathcal{L}_{\text{REJ}h^{(k)}}(\tilde{x}, y_j)$$

17:         **end if**
        {Select whether to perturb by comparing success rates against past models for the clean and perturbed samples.}
18:

$$\text{err}_{\text{clean}} = \frac{1}{i} \sum_{0 \leq k \leq i} \mathbb{1} \left[ F\left(h^{(k)}\right)(x_j) \neq y_j \right]$$

19:

$$\text{err}_{\text{perturbed}} = \frac{1}{i} \sum_{0 \leq k \leq i} \mathbb{1} \left[ F\left(h^{(k)}\right)(\tilde{x}_j) \notin \{y_j, \perp\} \right]$$

        {Do not perturb if the perturbation reduces robust rejection accuracy less on average than leaving the points unchanged.}
20:         **if** $\text{err}_{\text{perturbed}} < \text{err}_{\text{clean}}$ **then**
21:             $\tilde{x}_j = x_j$
22:         **end if**
23:         $E'_j = \tilde{x}_j, y_i$
24:     **end for**
25: **end for**
26: **Return:** $\hat{E}$

---

Given a sample $(x, y)$, the attacker's goal is to flip the label, and, simultaneously, to avoid rejection; hence, we maximize the following loss:

$$\mathcal{L}_{\text{DISC}}(x, y) = \mathcal{L}_{\text{CE}}(h_c^s(x), y) + \lambda \mathcal{L}_{\text{CE}}(h_d^s(x), 1)$$

where class 1 for $h_d$ corresponds to test data, signalling rejection, and where $h^s$ returns the softmax activations of $h$. Maximizing $\mathcal{L}_{\text{DISC}}$ then minimizes the confidence in the true label and the probability of rejection.

Figures 2 and 3 show our adaptive attack's performance on MNIST and CIFAR-10. $\tau$ is a key hyperparameter of Rejectron, which determines the confidence needed by $h_d$ to reject a sample; to evaluate Rejectron fairly, we report the results on best-performing value of $\tau$, based on (transductive) robust rejection accuracy; see Table 5. On CIFAR-10, performance is near-zero and rejection rate is near 100% for small values of $\tau$. The best-performing value of $\tau$ is 1 (effectively eliminating the

---

**Algorithm 2** TRANSDUCTIVE REJECTION-AWARE TRANSFER

---

**Require:** A model $h$, a clean labelled test point $(x, y)$, an adversarial budget of $\epsilon$, and a radius used for rejection $\epsilon_{\text{defense}}$.

{Search for a perturbation $\tilde{x}$ of $x$ for which $h$ predicts $\hat{y} \neq y$ robustly.}

1:

$$\tilde{x} = \arg \max_{\|\tilde{x}-x\| \leq \epsilon} \left[ \mathcal{L}_{\text{CE}}(h^s(\tilde{x}), y) + \lambda \left\| \tilde{x} - \arg \max_{\|x'-\tilde{x}\| \leq \epsilon_{\text{defense}}} \mathcal{L}_{\text{DB},h}(x') \right\|, \right]$$

where $\mathcal{L}_{\text{CE}}$ is the cross-entropy loss, $h^s$ returns the softmax activations of $h$ and where $\mathcal{L}_{\text{DB},h}(x) = \text{rank}_2 h^s(x) - \max h^s(x)$.

{If the attack did not succeed against $h$ (in other words, if $h$ does not robustly predict $\hat{y} \neq y$), check whether to leave $x$ unperturbed.}

2:

$$x' = \arg \max_{\|x'-\tilde{x}\| \leq \epsilon_{\text{defense}}} \mathcal{L}_{\text{CE}}(h^s(x'), h(\tilde{x}))$$

3: **if** $h(x') \neq h(\tilde{x}) \vee h(\tilde{x}) = y$ **then**
4:     Leave $x$ unperturbed if $F(h)$ rejects it, or if $h(x) \neq y$.
5:

$$x'' = \arg \max_{\|x''-x\| \leq \epsilon_{\text{defense}}} \mathcal{L}_{\text{CE}}(h^s(x''), h(x))$$

6:     **if** $h(x) \neq y \vee h(x'') \neq h(x)$ **then**
7:         $\tilde{x} = x$
8:     **end if**
9: **end if**
10: **Return:** $\tilde{x}$

---

**Algorithm 3** INDUCTIVE REJECTION-AWARE ATTACK

---

**Require:** A model $h$, and a clean labelled test point $(x, y)$, an adversarial budget of $\epsilon$, and a radius used for rejection $\epsilon_{\text{defense}}$.

1: Search for a perturbation $\tilde{x}$ of $x$ for which $h$ predicts $\hat{y} \neq y$ robustly.

$$\tilde{x} = \arg \max_{\|\tilde{x}-x\| \leq \epsilon} \left[ \mathcal{L}_{\text{CE}}(h^s(\tilde{x}), y) + \lambda \left\| \tilde{x} - \arg \max_{\|x'-\tilde{x}\| \leq \epsilon_{\text{defense}}} \mathcal{L}_{\text{DB},h}(x') \right\| \right]$$

where $\mathcal{L}_{\text{CE}}$ is the cross-entropy loss, $h^s$ returns the softmax activations of $h$ and where $\mathcal{L}_{\text{DB},h}(x') = \text{rank}_2 h^s(x') - \max h^s(x')$

2: **Return:** $\tilde{x}$

---

possibility of rejection), leading to a rejection rate of 0; this behavior on CIFAR-10 illustrates the algorithm's struggles with the practical high-complexity deep learning setting.

## C   ADDITIONAL EXPERIMENTS

### C.1   WARM START IN TLDR

| Warm start (epochs) | Rejection Rate | Robust Rejection Accuracy |
|---|---|---|
| 0 | 0.813 | 0.153 |
| **500** | **0.531** | **0.177** |
| 1000 | 0.830 | 0.171 |

Here we perform experiments showing that in training TLDR, it is best to first trains a baseline model without transductive regularization $L_{\text{test}}$ in the early stage (warm start) and then add transductive regularization for later training.

We generate the data with 100 Gaussians (one per class) equally spaced in $l_\infty$ with a separation of 3 units between means. The adversarial budget is 2 units, and we ensure that the data is sparse by generating 10 samples per class. The models are 10 layer feedforward networks with skip connections.

The synthetic models are trained for 1000 epochs total; we see the best performance when the model has transductive regularization but is allowed to learn an initial baseline model before transductive regularization is used in training. Doing so reduces the risk of the regularization term harming performance.

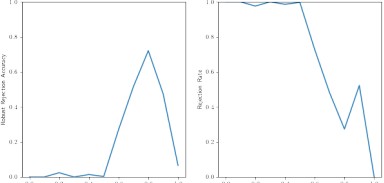 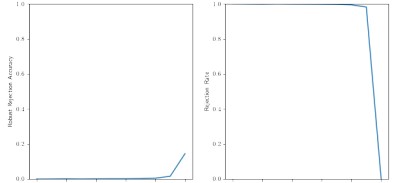

Figure 2: Effects of $\tau$ on performance of Rejectron on MNIST with attacker GMSA ($\mathcal{L}_{\text{DISC}}$).

Figure 3: Effects of $\tau$ on performance of Rejectron on CIFAR-10 with attacker GMSA ($\mathcal{L}_{\text{DISC}}$).

## C.2 GMSA METHOD

| TLDR Components | | Attacker | MNIST | | CIFAR-10 | |
|---|---|---|---|---|---|---|
| Rejection | Transductive Regularization | | $p_{\text{REJ}}$ | Robust accuracy | $p_{\text{REJ}}$ | Robust accuracy |
| ✓ | ✓ | GMSA$_{\text{AVG}}$ ($\mathcal{L}_{\text{REJ}}$) | 0.796 | 0.968 | 0.195 | 0.744 |
| ✓ | ✓ | GMSA$_{\text{MIN}}$ ($\mathcal{L}_{\text{REJ}}$) | 0.588 | 0.967 | 0.208 | 0.739 |
| ✓ | ✗ | GMSA$_{\text{AVG}}$ ($\mathcal{L}_{\text{REJ}}$) | 0.646 | 0.975 | 0.179 | 0.725 |
| ✓ | ✗ | GMSA$_{\text{MIN}}$ ($\mathcal{L}_{\text{REJ}}$) | 0.202 | 0.980 | 0.182 | 0.733 |
| ✗ | ✓ | GMSA$_{\text{AVG}}$ ($\mathcal{L}_{\text{CE}}$) | – | 0.900 | – | 0.516 |
| ✗ | ✓ | GMSA$_{\text{MIN}}$ ($\mathcal{L}_{\text{CE}}$) | – | 0.914 | – | 0.601 |
| ✗ | ✗ | GMSA$_{\text{AVG}}$ ($\mathcal{L}_{\text{CE}}$) | – | 0.935 | – | 0.516 |
| ✗ | ✗ | GMSA$_{\text{MIN}}$ ($\mathcal{L}_{\text{CE}}$) | – | 0.942 | – | 0.556 |

Table 8: Full ablation results of TLDR.

We present extended results of our defense ablation and compare the results of GMSA$_{\text{AVG}}$, which optimizes the average loss of past iterations, and GMSA$_{\text{MIN}}$, which optimizes the worst-case loss. See (Chen et al., 2022). We can see that while the two perform about the same on the full TLDR defense (GMSA$_{\text{MIN}}$ performs slightly better), GMSA$_{\text{AVG}}$ is much stronger for models not incorporating both components.

## C.3 REJECTION RADIUS

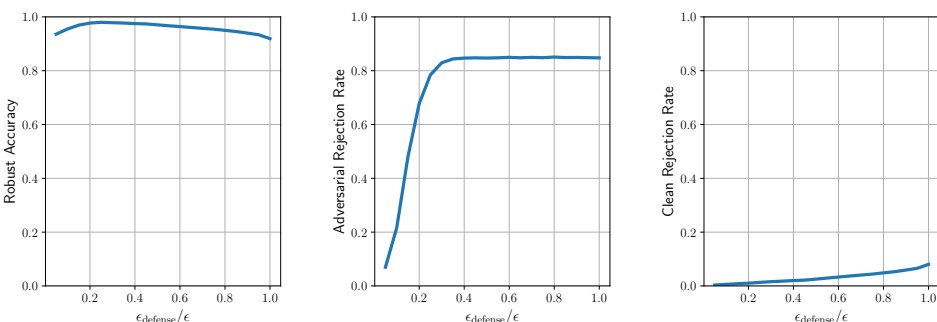

Figure 4: Effects of rejection radius $\epsilon_{\text{defense}}$ on MNIST (inductive) with attacker PGD ($\mathcal{L}_{\text{REJ}}$).

The rejection radius $\epsilon_{\text{defense}}$ is an important hyper-parameter for TLDR; however, the model's performance is not very sensitive to it. Figure 4 shows the trend of robust accuracy, the rejection rate on adversarial test data, and the rejection rate on clean test data, for the inductive classifier on MNIST; Figure 5 shows those for TLDR. The robust accuracy remains stable. The theoretical analysis suggests setting the radius to $\epsilon/3$ where $\epsilon$ is the adversarial budget. Given TLDR's low sensitivity to the parameter, we use $\epsilon/4$ for consistency as the inductive case performs best with that setting. The rejection rate on the adversarial test data rises rapidly with the rejection radius (reaching 0.949 for

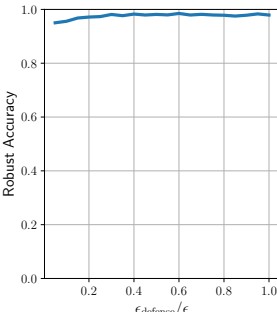 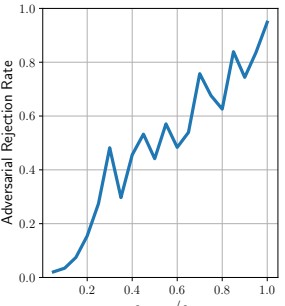 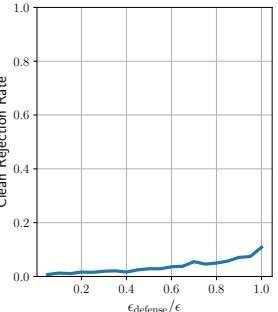

Figure 5: Effects of rejection radius $\epsilon_{\text{defense}}$ on MNIST (TLDR) with attacker GMSA ($\mathcal{L}_{\text{REJ}}$).

TLDR for $\epsilon_{\text{defense}} = \epsilon$), but the rejection rate on clean data increases much more slowly (0.108 when $\epsilon_{\text{defense}} = \epsilon$). So among all rejected inputs only a few are clean inputs, leading to low errors as desired.

The rejection rate on clean inputs is presented for the transductive case in order to illustrate the difference in effects on clean and perturbed data, but, as the adversary may select to perturb, some clean points were not in the training set, and, hence, the clean rejection rates should not be considered reliable. The rejection rates rise with the rejection radius: adversarial rejection rates increase rapidly as the rejection radius increases, while clean rejection rates increase only slowly. In all cases, far more perturbed samples are rejected than clean samples.

### C.4 BINARIZATION TEST ON PGD ($\mathcal{L}_{\text{REJ}}$)

| | MNIST | | | | CIFAR-10 | | | |
|---|---|---|---|---|---|---|---|---|
| Decision Boundary Closeness | ASR | RASR | Inverted ASR | Inverted RASR | ASR | RASR | Inverted ASR | Inverted RASR |
| 0.9 | 0.935 | 0.451 | 1.0 | 0.375 | 0.973 | 0.824 | 0.971 | 0.781 |
| 0.999 | 0.945 | 0.394 | 1.0 | 0.447 | 0.976 | 0.813 | 0.964 | 0.790 |
| 0.99999 | 0.953 | 0.414 | 0.981 | 0.434 | 0.974 | 0.819 | 0.938 | 0.813 |

Table 9: Results of the binarization test applied to PGD ($\mathcal{L}_{\text{REJ}}$).

Finally, to evaluate the effectiveness with which $\mathcal{L}_{\text{REJ}}$ targets rejection, we apply the binarization test (Zimmermann et al., 2022). As the binarization test is designed for inductive defenses we evaluate on PGD ($\mathcal{L}_{\text{REJ}}$), and as the binarization test assumes that rejection does not depend on the generated dataset or the modified model, we modified $\mathcal{L}_{\text{REJ}}$ to target the original model in the calcuation of $\mathcal{L}_{\text{DB},h}$ (e.g. we wish to avoid rejection as if the model was unchanged).

For the inverted case, we modified $\lambda'$, setting it to -10 (we are seeking rejection, not avoiding it). As noted in Appendix C , we drop the rejection-avoidance term when $h(\tilde{x}) = y$; hence, the negated second term poses issues for maximization in PGD (e.g. PGD would preferentially select perturbations which do not succeed). To avoid this issue, we have added an additional success indicator to our attack objective, which we use to ensure that PGD selects the loss-maximizing successful perturbation. Without these modifications, we observed low attack success rates in the inverted test; however, the results with these simple changes do indicate that our attack does take the rejection component of the defense into account, the key purpose of the inverted test.

The attack settings for the regular test are unchanged from those used for evaluation. For the test settings, we chose values as close as possible to those used in (Zimmermann et al., 2022), with a single boundary sample, with 200 samples sampled from each of the surfaces and corners of the $l_\infty$ ball, with 512 trials per experiment. We used 81 inner samples for MNIST, and 253 inner samples for CIFAR-10, selected to maximize subject to the requirement that the total sample count is below the dimensionality of the features. In both cases, the base model is a standard adversarially trained model trained on that dataset, transformed into a selective classifier with the transformation F.

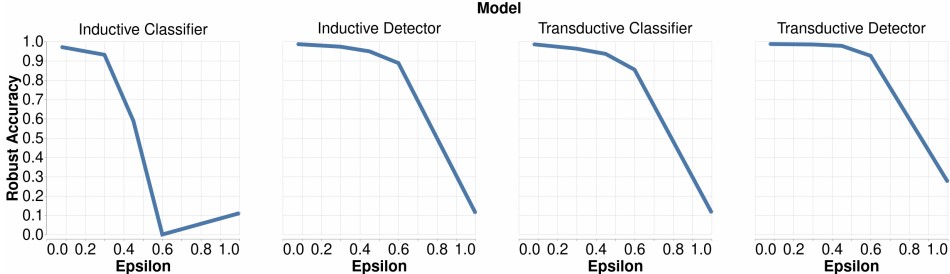

Figure 6: Robustness scaling with adversarial budget $\epsilon$ on MNIST

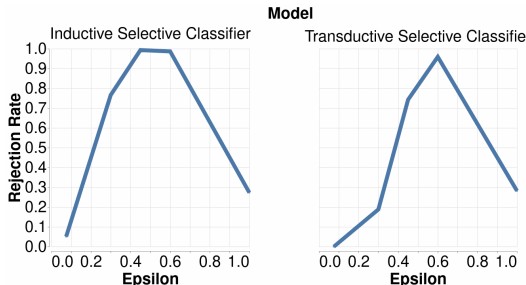

Figure 7: Rejection rate scaling with adversarial budget $\epsilon$ on MNIST.

We ran the test for a range of values for the decision boundary closeness, a hyperparameter determining the test hardness. ASR is the rate at which the attack successfuly found a perturbation which both flips the label and evades detection; RASR is the maximum of the success rates on surfaces and corners. While the ASR values in some experiments are slightly below the cutoff of 0.95 and are technically failures, they do indicate that the attack is successfully targeting the defense. While a slightly stronger attack may exist, these results do not indicate significant unreliability in our evaluation of the robustness of TLDR.

### C.5    ABLATION ON ATTACKS: ATTACK RADIUS

The theory suggests that incorporating rejection can allow a transductive learner to tolerate perturbations twice as large; we investigate how transduction and rejection affects the robustness as $\epsilon$ grows (models are adversarially trained with the corresponding $\epsilon$ and the selective classifiers use a rejection radius of $\epsilon/2$). The results are shown for the natural choice of adversary, as in the experiment section (e.g. GMSA with $\mathcal{L}_{\text{REJ}}$ for the transduction+rejection). For selective classifiers, the rejection rate scaling is shown.

We see that the combination of rejection and transduction does indeed maintain high accuracy for larger $\epsilon$; at $\epsilon = 0.6$, it has 96.2% of the robust accuracy that transduction alone had for $\epsilon = 0.3$. This aligns with the theory, given the increased constant factors of $\text{OPT}_{\mathcal{U}^2}$ in Corollary A.13 compared to the results for classifiers in (Montasser et al., 2021).

Note also the behavior of the inductive classifier: accuracy improves past $\epsilon = 0.6$. To see why, note that a model adversarially trained for $\epsilon \geq 1$ will return near-uniform predictions for all classes (resulting in a robust accuracy of approximately 10%, as seen), making finding adversarial examples slightly more difficult than for smaller $\epsilon$ where this does not occur. The decline in rejection rate for very large $\epsilon$ is a similar phenomenon.

### C.6    WEIGHTING OF $\mathcal{L}_{\text{REJ}}$

We examine the effect of the hyperparameter $\lambda'$ between the cross-entropy and rejection-avoidance terms in $\mathcal{L}_{\text{REJ}}$ on MNIST; see Equation 3. In the inductive case, as shown in Figure 8, there is little sensitivity to $\lambda'$ in either attack success rate or rejection rate. When targeting TLDR, there is little

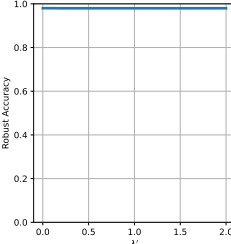 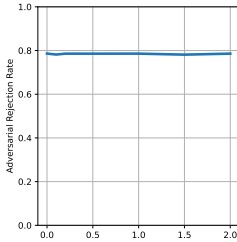

Figure 8: Effects of $\lambda'$ on results of PGD optimizing $\mathcal{L}_{\text{REJ}}$ targeting adversarial training with rejection on MNIST.

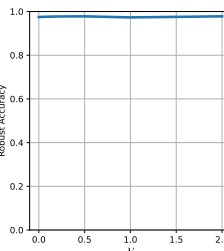 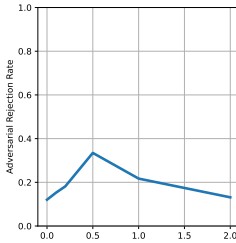

Figure 9: Effects of $\lambda'$ on results of GMSA optimizing $\mathcal{L}_{\text{REJ}}$ targeting TLDR on MNIST.

sensitivity in terms of attack success rate as seen in Figure 9; rejection rate is highest for intermediate values of $\lambda'$ but, as expected, rejection rate declines with $\lambda'$ beyond that.

### C.7 ROBUSTNESS TO $l_2$

| Setting | Defense | Attacker | MNIST | | CIFAR-10 | |
|---------|---------|----------|-------|---------------|----------|---------------|
| | | | $p_{\text{REJ}}$ | Robust accuracy | $p_{\text{REJ}}$ | Robust accuracy |
| Induction | AT (Madry et al., 2018) | AutoAttack | – | 0 | – | 0.445 |
| Rejection only | AT (with rejection) | PGD ($\mathcal{L}_{\text{REJ}}$) | 0.112 | 0.921 | 0.130 | 0.754 |
| Transduction only | TADV (Chen et al., 2022) | GMSA ($\mathcal{L}_{\text{CE}}$) | – | 0.913 | – | 0.813 |
| Transduction+Rejection | **TLDR (ours)** | GMSA ($\mathcal{L}_{\text{REJ}}$) | **0.078** | **0.933** | **0.007** | **0.845** |

Table 10: Results on MNIST and CIFAR-10 up to $l_2$ budget. The strongest attack against each defense is shown. The best result is **boldfaced**.

To evaluate our defense's generality, we consider robustness to $l_2$ as well and compare to the strongest defenses from each setting in Table 10; on MNIST we use $\epsilon = 5$ and on CIFAR-10 we use $\epsilon = 128/255$. We observe strong performance from TLDR, outperforming defenses with transduction or rejection alone.

### C.8 GENERALIZATION OF TLDR

To evaluate how closely TLDR's generalization follows the our provided bounds in Theorems A.9 and A.12, we apply TLDR to randomly-sampled subsets of the MNIST training and test sets. In each case, we run ten trials and present the robust error (1 - robust accuracy) with attacker GMSA ($\mathcal{L}_{\text{REJ}}$). Given the large VC dimension of the model considered (LeNet) (Bartlett et al., 2017), the results shown are consistent with Theorem A.12; we wish to determine whether the actual errors observed follow the inverse-square relationship of the theorem.

In Figure 10, the size of the training set is set equal to the size of the test set (the standard assumption for our results); in Figure 11, the full training set is used and only the test set size is changed. See Appendix A.5 for a discussion of generalization bounds for train and test sets of differing sizes.

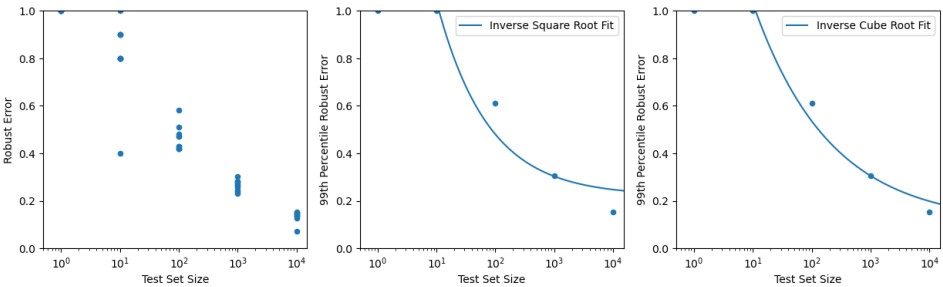

Figure 10: Generalization of TLDR with equal train and test size on MNIST.

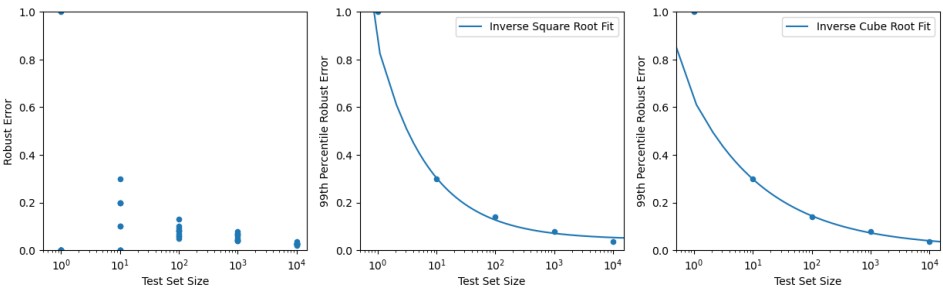

Figure 11: Generalization of TLDR with full training set on MNIST.

As the bounds are in PAC form, we use an estimate of the 99th percentile of error in order to evaluate the generalization of TLDR; these are calculated with a best-fit beta distribution of the results on each instance size.

We then consider the inverse-square-root fit of these 99th percentile error estimates; as the gurarantee takes the form of an upper bound, and error is upper bounded by 1, we exclude any error values equal to 1 (corresponding to instances where all trials had a robust accuracy of 0). We find that in the case where train size is fixed, the 99th percentile errors closely follow the inverse-square-root trend in the test set size $m$; while the results for equal train and test set sizes more closely follow an inverse-cube-root relationship in $m$.

# D    LIMITATIONS

While our framework is theoretical-sound with lower sampled complexity than the rejection-only case and with more relaxed optimality condition than the transductive-only case, our sample complexity proof under the transductive rejection case requires the non-emptiness of $\Delta$ in Theorem 4.1. While weaker conditions don't guarantee that we find a model satisfying the conditions, the result demonstrate that empirical defense incorporating both transduction and rejection have the potential to outperform others. Our proposed defense algorithm TLDR, though effective at improving the robust accuracy under rejection, incurs a high computational cost relative to standard adversarial training due to the joint training with the unlabeled data. If it is possible to delay evaluation until a sufficiently large batch of samples arrives, the cost can be made insignificant via amortization. The need to perform a full training process prior to evaluation means, however, that the defense is not suitable for latency-sensitive applications. Our adaptive attack is even more costly, as effectively attacking this defense using GMSA requires multiple iterations of the full transductive training process; hence, adversaries attacking TLDR require substantial resources.

