# OpenReview forum: "Two Heads are Better than One: Towards Better Adversarial Robustness by Combining Transduction and Rejection"
_ICLR.cc/2024/Conference — Submitted to ICLR 2024_

### Official Review · Reviewer_xAQ7 · 2023-10-29

**Soundness:** 3 good
**Presentation:** 3 good
**Contribution:** 3 good
**Rating:** 6
**Confidence:** 3

**Summary:**

This paper considers adversarial robustness in the transductive setting along with a rejection option. Previous papers only consider either the transductive setting or the inductive setting with a rejection option (but not both). This paper exploits the benefits of both settings to have a better theoretical sample complexity and better empirical robustness (than comparing papers). Moreover, the paper proposes a robust learning algorithm and an attack algorithm for the new transductive setting along with a rejection. The efficacy of the defense and attack algorithms are demonstrated on MNIST and CIFAR-10 (along with multiple baselines).

**Strengths:**

Originality: this paper clearly contrasts its problem setting (i.e., transductive+rejection) to existing settings (transductive only and rejection only).

Quality:  The main claim is the efficacy of considering two settings (i.e., transductive and rejection) at once and it is well supported by the theory (e.g., Theorem 4.1) and empirical evaluations.

Clarify: this paper is well-written. But, in Section 4, I think it would be better if this paper can show error bounds for the other two baseline settings.

Significance: This new view on settings on adversarial robustness could encourage some interesting ideas. However, I felt the transductive setup is too easy to achieve robustness by using the current achievement on adversarial robustness. See Weaknesses.

**Weaknesses:**

I overall like this paper, and the following includes a few remaining concerns.

One main concern is that it is possible that proposing learning methods for a robust classifier in the transductive setting with a rejection option is not novel (even though rigorously showing that this problem is easy as in Theorem 4.1 Is very meaningful).
First, I think by exploiting certified robustness methods (e.g., randomized smoothing) in the inductive setting, we can achieve 100% robustness in the transductive setting. In particular, in the transductive setting, it is fine to achieve robustness on a given test set. Thus, as long as we can find an upper bound of max in (2) (which can be done via known certified robustness methods), this can be leveraged to choosing a threshold for the selective classifier, which provides 100% robustness accuracy. Then, the remaining issue is how to reduce the false rejection rate, which is equivalent to making the upper bound tighter. In this regard, proposing a method to learn a robust classifier in the transductive setting with a rejection option may not add any novelty to advance the adversarial robustness research. I want to hear any opinion that considering transductive+rejection can potentially trigger to introduce interesting methods to advance adversarial robustness.

In the following, I have minor editorial comments.
* Page 3, “Rejection.”: \mathbb{R} is not defined.
* Page 5, “Proof Sketch”: R is not defined.
* Section 4. It would be better to show bounds for inductive-only and transductive-only to clearly contrast the benefit of Theorem 4.1.

**Questions:**

As mentioned in Weaknesses, proposing learning methods for a robust classifier in the transductive setting with a rejection option is not novel (as we can simply leverage certified robustness techniques in the inductive setting). By considering the transductive+rejection setting, are there any potential benefits for the adversarial learning communities to come up with new and interesting methods? Or, advancing methods in the inductive setting naturally advances methods in the transductive+rejection setting?

---

> ### Author Response · Authors · 2023-11-16
>
> Dear Reviewer xAQ7,
>
> Thank you for your valuable feedback and suggestions. We address your questions and concerns in detail below.
>
> > One main concern is that it is possible that proposing learning methods for a robust classifier in the transductive setting with a rejection option is not novel (even though rigorously showing that this problem is easy as in Theorem 4.1 Is very meaningful). First, I think by exploiting certified robustness methods (e.g., randomized smoothing) in the inductive setting, we can achieve 100% robustness in the transductive setting. In particular, in the transductive setting, it is fine to achieve robustness on a given test set. Thus, as long as we can find an upper bound of max in (2) (which can be done via known certified robustness methods), this can be leveraged to choosing a threshold for the selective classifier, which provides 100% robustness accuracy. Then, the remaining issue is how to reduce the false rejection rate, which is equivalent to making the upper bound tighter.
>
> We assume that the suggested approach is to generate certificates for each test point, and select a minimum certificate radius such that the rejection rate is some fixed value; do we understand the proposed approach correctly? While this can be used to bound rejection rate, it does not necessarily increase robust accuracy. Similarly, the $\sigma$ parameter for randomized smoothing can be selected to ensure some minimum degree of robustness for all points; however, this likewise has limitations. In particular, this trades off accuracy for robustness [1], and ensuring robustness beyond the rejection radius for all points eliminates any benefit from rejection.
>
> [1] Jeremy M Cohen, Elan Rosenfeld, and J. Zico Kolter. Certified adversarial robustness via randomized smoothing, 2019.
>
> > In this regard, proposing a method to learn a robust classifier in the transductive setting with a rejection option may not add any novelty to advance the adversarial robustness research. I want to hear any opinion that considering transductive+rejection can potentially trigger to introduce interesting methods to advance adversarial robustness.
>
> We have already shown several promising results illustrating the potential of transduction+rejection; [2] provides another compelling illustration of the potential of the setting by showing that the combination of transduction and rejection enables capabilities (e.g. robustness to arbitrary perturbations) not feasible in any other setting. There exist many potential future directions for work in this direction, for one example, we believe that the development of defenses with transduction+rejection leveraging more sophisticated semi-supervised learning techniques may enable substantial further improvements to robustness.
>
> [2] Shafi Goldwasser, Adam Tauman Kalai, Yael Kalai, and Omar Montasser. Beyond perturbations: Learning guarantees with arbitrary adversarial test examples, 2020.
>
> > As mentioned in Weaknesses, proposing learning methods for a robust classifier in the transductive setting with a rejection option is not novel (as we can simply leverage certified robustness techniques in the inductive setting). By considering the transductive+rejection setting, are there any potential benefits for the adversarial learning communities to come up with new and interesting methods? Or, advancing methods in the inductive setting naturally advances methods in the transductive+rejection setting?
>
> As noted above, this setting presents numerous opportunities for the adversarial learning community. In addition, certain defenses with transduction+rejection (including TLDR) do benefit directly from advancement in the inductive setting; in the case of TLDR this is due to the internal application of inductive attacks (PGD) in the training losses as well as during the rejection step; improved internal attacks will improve the performance of TLDR.

---

### Official Review · Reviewer_XrZg · 2023-10-31

**Soundness:** 3 good
**Presentation:** 3 good
**Contribution:** 3 good
**Rating:** 6
**Confidence:** 4

**Summary:**

The paper considers a transductive model where the learner receives an iid labeled training set (drawn from some unknown distribution) and an unlabeled test set of examples. The test examples are either: (a) iid from the same distribution from which the training examples are drawn, or (b) adversarial corruptions according to a predefined perturbation set U (e.g. lp ball). If a test example is of type (a), the learner should label it correctly, but if it is of type (b) the learner should either label it correctly or abstain. That is, in this tansductive model mistakes and abstentions on iid examples are penalized, and only mistakes on adversarial examples are penalized (learner can abstain).

The model proposed and studied in this paper can be viewed as:
(1). a restriction of the model studied in (Goldwasser et al., 2020), in the sense that in this paper the adversary is limited and can only perturb according to a predefined perturbation set U. On the other hand, this paper obtains better error+abstention rates as in Table 1.

(2). a relaxation of the transductive model studied in (Montasser et al., 2021), where this paper allows the learner to abstain.

The paper contributes theoretical learning guarantees based on the VC dimension. Additionally, the paper conducts empirical experiments on MNIST and CIFAR10 to evaluate transductive selective classification for adversarial robustness.

**Strengths:**

The paper contributes good theoretical results that fill some gaps in the literature. The paper does a good job in putting their results/guarantees in the context of prior work, and explaining what's different from prior work.
Additionally, it is nice to have empirical experiments supplementing the theory.

The paper is well-written and easy to read.

**Weaknesses:**

About the theoretical results:
To some extent, I think the paper is incremental. In the sense that both the model studied and the techniques used are a combination of previous models and techniques used in prior work (Montasser et al., 2021, Goldwasser et al., 2020, Tramer 2022). Moreover, the quantitative sample complexity bounds obtained in this paper exactly match the bounds obtained in Montasser et al., 2021, with the only difference being an improvement in the soundness condition as indicated in Table 1. Which translates to an improvement on the distributional assumption of "robust realizability" OPT_{U^{2/3}} = 0 (vs. OPT_{U^2}=0). This is an interesting difference from the prior results of (Montasser et al., 2021), but it is not clear from this paper whether this difference is inherent / fundamental, or merely just a limitation of the techniques.

In fact, the work of Tramer 2022, suggests that in the inductive setting, there is an equivalence between the problem of robustly classifying adversarial examples and the problem of abstaining on adversarial examples (with some difference in radius of perturbation sets). And the results in this paper would be more convincing/stronger if the paper proves/disproves whether a similar equivalence holds in the transductive setting.

**Questions:**

For suggestions, see "weaknesses" section.

---

> ### Author Response · Authors · 2023-11-16
>
> Dear Reviewer XrZg,
>
> Thank you for your valuable feedback and suggestions. We address your questions and concerns in detail below.
>
> > About the theoretical results: To some extent, I think the paper is incremental. In the sense that both the model studied and the techniques used are a combination of previous models and techniques used in prior work (Montasser et al., 2021, Goldwasser et al., 2020, Tramer 2022). Moreover, the quantitative sample complexity bounds obtained in this paper exactly match the bounds obtained in Montasser et al., 2021, with the only difference being an improvement in the soundness condition as indicated in Table 1. Which translates to an improvement on the distributional assumption of "robust realizability" OPT_{U^{2/3}} = 0 (vs. OPT_{U^2}=0). This is an interesting difference from the prior results of (Montasser et al., 2021), but it is not clear from this paper whether this difference is inherent / fundamental, or merely just a limitation of the techniques.
>
> As discussed in the remark following the proof sketch for Theorem 4.1, the approach is not a simple combination of these techniques; more direct approaches fail to derive any improvement. We show a separation result in Theorem A.11, which shows that there exists a distribution on which our bound holds with completeness, but where *no* learner with transduction alone can have asymptotic error below ${1\over 2}$.
>
> > In fact, the work of Tramer 2022, suggests that in the inductive setting, there is an equivalence between the problem of robustly classifying adversarial examples and the problem of abstaining on adversarial examples (with some difference in radius of perturbation sets). And the results in this paper would be more convincing/stronger if the paper proves/disproves whether a similar equivalence holds in the transductive setting.
>
> The argument of Theorem 4.1 can be adapted to show one direction of such an equivalence, in particular that the existence of a complete transductive learner with smaller radius implies the existence of a sound transductive learner with rejection with a larger radius. We leave the converse open, however, we conjecture that it does not hold; in particular, it is unclear how to reduce a complete learner to a sound one.

---

### Official Review · Reviewer_qXLN · 2023-11-04

**Soundness:** 3 good
**Presentation:** 3 good
**Contribution:** 2 fair
**Rating:** 6
**Confidence:** 3

**Summary:**

This paper addresses the problem of defending against adversarial perturbations with the techniques of "transduction" and "rejection." Transduction involves training on the set of test data, while rejection involves refusing to make predictions when the input is uncertain. The paper discusses previous works that have explored these techniques and their limitations.

The contribution of the paper is mainly on improving the generalization bound of adversarial learning with two options *rejection* and *transduction*, over previous work Tramer et al. and  Goldwasser et al.2021. The paper also conducts experiments to demonstrate the efficacy of rejection and transduction.

**Strengths:**

Theoretical Foundation: The paper provides a strong theoretical foundation for its proposed approach by developing the upper bound of generalization error, which can help improve the understanding of the problem.

Practical Experiments: The inclusion of extensive experiments using state-of-the-art attacks strengthens the empirical support for the proposed solutions.

**Weaknesses:**

From the result in Peng et al.2023, data (diffusion augmented approach) is an essential factor to provide genuine boost of adversarial performance (over 10 points improvement).  Instead, introducing new training method (transduction) does not give good improvement, and may not contribute much to the ultimate solution of adversarial robustness. More importantly, it is hard to compare these results as the assumption of availability of test data is different and the robust accuracy has different meaning when using rejection operation.

Transduction requires training on test data (unlabeled), which is far from realistic. It is better and important to design the algorithm and develop the theory based on online assumption of test data,  i.e., each test data point is arriving sequentially.

**Questions:**

However, in Table 6 (Ablation Study), introducing L_test does not provide much improvement. Thus, the transduction is mainly for theoretical improvement. Is this correct?

**Details Of Ethics Concerns:**

N/A, Theoretical analysis on robust accuracy and no immediate ethical concerns.

---

> ### Author Response · Authors · 2023-11-16
>
> Dear Reviewer qXLN,
>
> Thank you for your valuable feedback and suggestions. We address your questions and concerns in detail below.
>
> > From the result in Peng et al.2023, data (diffusion augmented approach) is an essential factor to provide genuine boost of adversarial performance (over 10 points improvement). Instead, introducing new training method (transduction) does not give good improvement, and may not contribute much to the ultimate solution of adversarial robustness. More importantly, it is hard to compare these results as the assumption of availability of test data is different and the robust accuracy has different meaning when using rejection operation.
>
> We agree that additional data is beneficial, but this is not mutually exclusive with our contributions. Without any additional generated data, we significantly exceed the results of Peng et al. on CIFAR-10 (81.6% robust accuracy compared to 71.1%, with a greater relative improvement when equalizing architectures); we have additionally conducted experiments on CIFAR-100, on which TLDR continues to significantly outperform their results (57.9% compared to 38.7%).
>
> While these are indeed different notions of robust accuracy, it is necessary to compare between settings for our work, as our focus is on demonstrating the potential advantages of one setting (transduction+rejection) over others. While such comparisons are not direct, the definition of robust accuracy for each setting in all cases captures the same concept, the fraction of samples on which we are correct (including notions of error, such as rejecting clean samples, which capture the disadvantages of a setting). Hence, we demonstrate the potential increase in the fraction of samples on which correctness is possible when comparing between settings.
>
> > Transduction requires training on test data (unlabeled), which is far from realistic. It is better and important to design the algorithm and develop the theory based on online assumption of test data, i.e., each test data point is arriving sequentially.
>
> By retraining periodically, once a sufficiently large batch of samples arrives, transductive defenses can be easily adapted to a batched online setting, in particular for applications with minimal latency requirements. The results for the single sample/unbalanced online setting are in Appendix A.5, and suggest that transductive learners in general are suited to the batched rather than single-sample online setting, and as such this is not a significant limitation of our work.
>
> > However, in Table 6 (Ablation Study), introducing $l_\text{test}$ does not provide much improvement. Thus, the transduction is mainly for theoretical improvement. Is this correct?
>
> Note that all defenses shown in Table 6 are transductive; transduction in TLDR has two components, $l_\text{test}$ and private randomness. Note that $l_\text{test}$ has the additional benefit of substantially increasing the difficulty of the optimization problem to be solved by the adversary (instead of a simple maximization, the adversary must solve a maximin problem in the presence of $l_\text{test}$). Without private randomness or $l_\text{test}$, we reduce to the inductive case.

---

### Official Review · Reviewer_kR75 · 2023-11-06

**Soundness:** 3 good
**Presentation:** 3 good
**Contribution:** 3 good
**Rating:** 8
**Confidence:** 4

**Summary:**

The paper proposes a new methodology by combining rejection and transduction as two methods for augmenting the robustness of neural networks. The idea is based on a theoretical analysis such that if the test data points are too close to the decision boundary.  The method demonstrates a theoretical improvement compared to the previous methods due to this combination. The superiority of the method's performance is demonstrated by the extensive experiments on CIFAR10 and MNIST datasets.

**Strengths:**

- The paper starts with a rigorous analysis of the error-bound for a classifier with the rejection and transduction options. While both augmentations and their theoretical analysis are well-known in the literature, it is the first paper studying them jointly.

- The paper is generally well-written and offers good intuitions over the methodology and the theoretical study.

**Weaknesses:**

- Is it necessary in the proof of Theorem 4.1 to consider z' as $1/3$ distance from the actual perturbation? Is it just an arbitrary constant or it is necessary to use exactly $\frac{1}{3}$?

- The proposed bound depends on the complexity of the hypothesis set $H$. Given the assumption that it should contain a classifier with zero error, how large such a value (VC(H)) will be in practice?

- To validate the proposed bound, it is necessary to show that by increasing the unlabeled test size, the error decreases with high probability (according to the given bound). It would be nice if an experiment on the same dataset with a different number of test samples is added (performance as the function of the number of test samples). It can be done on multiple datasets with different complexities (CIFAR10, Imagenet) to demonstrate the performance of the method better.

**Questions:**

Please see the weaknesses part.

---

> ### Author Response · Authors · 2023-11-16
>
> Dear Reviewer kR75,
>
> Thank you for your valuable feedback and suggestions. We address your questions and concerns in detail below.
>
> > Is it necessary in the proof of Theorem 4.1 to consider z' as  $1\over 3$ distance from the actual perturbation? Is it just an arbitrary constant or it is necessary to use exactly $1\over 3$?
>
> It is not necessary to use exactly $1\over 3$; the argument can be adapted to handle $\tilde{z}’$ of varying distances from the perturbation. however, this provides the strongest results.
>
> To see this, consider $\tilde{z}’ = \tilde{x} + (\tilde{z}’ - \tilde{x}) \cdot p$ for $p \in [0,1]$. For the argument to hold, we require at least $p\epsilon$ robustness about $\tilde{z}’$ and nonempty intersection between a $p \epsilon$ ball about $\tilde{z}’$ and a $r$ ball about $\tilde{z}’$ where $r$ is the rejection radius. So $r \ge (1 - 2p)\epsilon$. But to avoid rejection on clean samples, we require at least $r$ robustness about $\tilde{z}’$; hence we require $\max\{ p\epsilon, (1-2p)\epsilon\}$ robustness about $\tilde{z}’$. Hence, soundness demands $\text{OPT}_{\max\{2p\epsilon, (1-p)\epsilon \} }$, minimized at $p ={1\over 3}$.
>
> > The proposed bound depends on the complexity of the hypothesis set $H$. Given the assumption that it should contain a classifier with zero error, how large such a value (VC(H)) will be in practice?
>
> This assumption is only necessary in the realizable case; we provide results for the agnostic case as well (see Appendix A.4). However, see [1] for results on VC-dimension for neural networks with ReLU activation; this suggests that the VC dimension of common models for image classification may be on the order of hundreds of millions.
>
> [1] Peter L. Bartlett, Nick Harvey, Christopher Liaw, Abbas Mehrabian. Nearly-tight VC-dimension and pseudodimension bounds for piecewise linear neural networks, 2017.
>
> > To validate the proposed bound, it is necessary to show that by increasing the unlabeled test size, the error decreases with high probability (according to the given bound). It would be nice if an experiment on the same dataset with a different number of test samples is added (performance as the function of the number of test samples). It can be done on multiple datasets with different complexities (CIFAR10, Imagenet) to demonstrate the performance of the method better.
>
> We are running these experiments; we will update when results are available.

---

> > ### Author Response · Authors · 2023-11-23
> >
> > We have added results on the generalization of TLDR on MNIST which show that, as expected, the error decreases with high probability as the test set size increases. This holds both in the case of equal training and test set sizes and in the case where the test set size is held fixed. See Appendix C.8 of the revision for the full details.

---

> > > ### Comment · Reviewer_kR75 · 2023-11-23
> > >
> > > I appreciate your changes in the new revision and your comprehensive response.

---

### Meta-Review · Area_Chair_qZDg · 2023-12-13

**Metareview:**

The paper considers a restriction of an adversarially robust prediction model studied in (Goldwasser et al., 2020), in which the learner when presented with samples at test time (without a label) can predict or abstain. The learner is penalized if it makes errors on unperturbed samples, or if it abstains on a sample that is unperturbed. Compared to  (Goldwasser et al., 2020), the adversary is only allowed some predefined perturbation set (e.g., $l_p$ ball), but the rates the paper gets are better. The main technical toolkit is to adapt a technique due to  Tramer for a classifier-to-detector reduction, along with techniques in (Montasser et al., 2021, Goldwasser et al., 2020).

On the positive side, the paper exposes tighter rates that can be gotten by combining techniques and model restrictions from prior works in the right way. On the other hand, as Reviewer XrZg pointed out, the paper largely combines techniques in (Montasser et al., 2021, Goldwasser et al., 2020, Tramer 2022). Overall, the results in the paper are tightenings of bounds in prior works, and essentially push to the limit prior techniques. The interest in the paper might be restricted to a fairly limited community.

**Justification For Why Not Higher Score:**

The techniques in the paper are largely adaptations of prior work. The improvements to the bounds themselves may be of interest only to a narrow community.

**Justification For Why Not Lower Score:**

N/A

---

### Decision · Program_Chairs · 2024-01-16

Reject